# EMERGENCE OF SHARED SENSORY-MOTOR GRAPHICAL LANGUAGE FROM VISUAL INPUT

## ABSTRACT

The framework of Language Games studies the emergence of languages in populations of agents. Recent contributions relying on deep learning methods focused on agents communicating via an idealized communication channel, where utterances produced by a speaker are directly perceived by a listener. This comes in contrast with human communication, which instead relies on a *sensory-motor channel*, where motor commands produced by the speaker (e.g. vocal or gestural articulators) result in sensory effects perceived by the listener (e.g. audio or visual). Here, we investigate if agents can evolve a shared language when equipped with a continuous sensory-motor system to produce and perceive signs, e.g. drawings. To this end, we introduce the Graphical Referential Game (GREG) where a speaker must produce a graphical utterance to name a visual referent object consisting of combinations of MNIST digits while a listener has to select the corresponding object among distractor referents, given the produced message. The utterances are drawing images produced using dynamical motor primitives combined with a sketching library. To tackle GREG we present CURVES: a multimodal contrastive deep learning mechanism that represents the energy (alignment) between named referents and utterances generated through gradient ascent on the learned energy landscape. We, then, present a set of experiments showing that our method allows the emergence of a shared, graphical language that generalizes to feature compositions never seen during training. We also propose a topographic metric to investigate the compositionality of emergent graphical symbols. Finally, we conduct an ablation study illustrating that sensory-motor constraints are required to yield interpretable lexicons.

## 1 INTRODUCTION

Understanding the emergence and evolution of human languages is a significant challenge that has involved many fields, from linguistics to developmental cognitive sciences (Christiansen & Kirby, 2003). Computational experimental semiotics (Galantucci & Garrod, 2011) has seen some success in modeling the formation of communication systems in populations of artificial agents (Cangelosi & Parisi, 2002; Kirby et al., 2014). More specifically, *Language Game* models (Steels & Loetzsch, 2012), have been used to show how a population of agents can self-organize a culturally shared lexicon without centralized coordination. Given the recent successes of artificial neural networks in solving complex tasks such as image classification (Krizhevsky et al., 2012; He et al., 2015; 2016; Dosovitskiy et al., 2021) and natural language understanding (Devlin et al., 2019; Radford et al., 2019; Brown et al., 2020), many works have leveraged them to study the emergence of communication in groups of agents (Lazaridou & Baroni, 2020), mainly using multi-agent deep reinforcement learning and language games (Nguyen et al., 2020; Mordatch & Abbeel, 2018; Lazaridou et al., 2018; Portelance et al., 2021; Chaabouni et al., 2021). These advances have made it possible to scale up language game models to environments where linguistic conventions are jointly learned with visual representations of raw image perception, as well as to environments where emergent communication is used as a tool to achieve joint cooperative tasks (Barde et al., 2022).

So far, most of these methods have considered only idealized symbolic communication channels based on discrete tokens (Lazaridou et al., 2017; Mordatch & Abbeel, 2018; Chaabouni et al., 2021) or fixed-size sequences of word tokens (Havrylov & Titov, 2017; Portelance et al., 2021). This predefined means of communication is motivated by language's discrete and compositional nature.

But how can this specific structure emerge during vocalization or drawing, for instance? Although fundamental in the investigation of the origin of language (Dessalles, 2000; Cheney & Seyfarth, 2005; Oller et al., 2019), this question seems to be neglected by recent approaches to Language Games (Moulin-Frier & Oudeyer, 2020). We, therefore, propose to study how communication could emerge between agents producing and perceiving continuous signals with a constrained *sensory-motor system*.

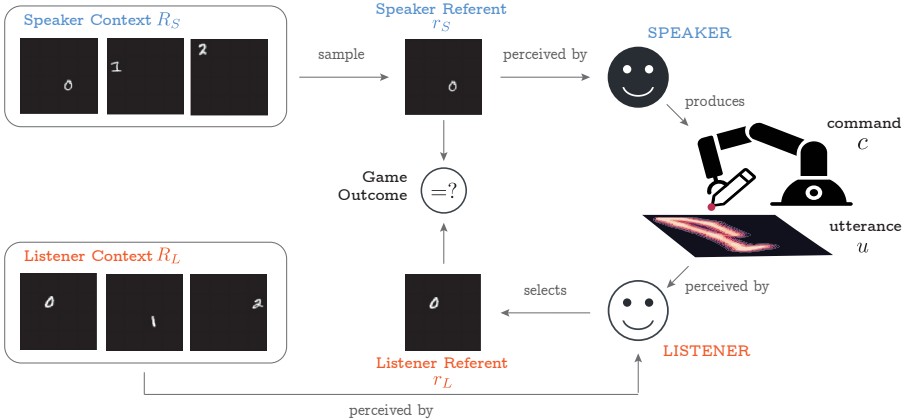

**Figure 1: The Graphical Referential Game:** During the game, the speaker's goal is to produce a motor command $c$ that will yield an utterance $u$ in order to denote a referent $r_S$ sampled from a context $\tilde{R}_S$. Following this step, the listener needs to interpret the utterance in order to guess the referent it denotes among a context $\tilde{R}_L$. The game is a success if the listener and the speaker agree on the referent ($r_L \equiv r_S$).

Such continuous constrained systems have been used in the cognitive science literature as models of sign production to study the self-organization of speech in artificial systems (de Boer, 2000; Oudeyer, 2006; Moulin-Frier et al., 2015). In this paper, we focus on a drawing sensory-motor system producing graphical signs. The sensory-motor system is made of Dynamical Motor Primitives (DMPs) (Schaal, 2006) combined with a sketching system (Mihai & Hare, 2021a) enabling the conversion of motor commands into images. Drawing systems have the advantage of producing 2D trajectories interpretable by humans while preserving the non-linear properties of speech models, which were shown to ease the discretization of the produced signals (Stevens, 1989; Moulin-Frier et al., 2015). We introduce the *Graphical Referential Game*: a variation of the original referential game, where a *Speaker* agent (top of Figure 1) has to produce a graphical *utterance* given a single target *referent* while a *Listener* agent (bottom of Figure 1) has to select an element among a context made of several referents, given the produced utterance (agents alternate their roles). In this setting, we first investigate whether a population of agents can converge on an efficient communication protocol to solve the graphical language game. Then, we evaluate the coherence and compositional properties of the emergent language, since it is one of the main characteristics of human languages.

Early language game implementations (Steels, 1995; 2001) achieve communication convergence by using contrastive methods to update association tables between object referents and utterances. While recent works use deep learning methods to target high-dimensional signals they do not explore contrastive approaches. Instead, they model interactions as a multi-agent reinforcement learning problem where utterances are actions, and agents are optimized with policy gradients, using the outcomes of the games as the reward signal (Lazaridou et al., 2017). In the meantime, recent models leveraging contrastive multimodal mechanisms such as CLIP (Radford et al., 2021) have achieved impressive results in modeling associations between images and texts. Combined with efficient generative methods (Ramesh et al., 2021), they can compose textual elements that are reflected in image form as the composition of their associated visual concepts. Inspired by these techniques, we propose CURVES: Contrastive Utterance-Referent associatiVE Scoring, an algorithmic solution to the graphical referential game. CURVES relies on two mechanisms: 1) The contrastive learning of an energy landscape representing the alignment between utterances and referents and 2) the generation of utterances that maximize the energy for a given target referent. We evaluate CURVES in two instantiations of the graphical referential game: one with symbolic referents encoded by one-hot vectors and another with visual referents derived from the multiple MNIST digits (LeCun et al.,

1998). We show that CURVES converges to a shared graphical language that enables a population of agents not only to name complex visual referents but also to name new referent compositions that were never encountered during training.

**Scope.** The idea of using a sensory-motor system to study the emergence of forms of combinatoriality in language dates back to methods investigating the origins of digital vocalization systems (de Boer, 2000; Oudeyer, 2005; Zuidema & De Boer, 2009). Such studies were conducted in the context of imitation games at the level of phonemes to observe the formation of speech utterances (syllables, words) that were systematically composed from lower-level meaningless elements (phonemes). This corresponded to the first level of compositionality within the notion of duality of patterning (Hockett & Hockett, 1960). Yet, these works did not consider referential games and did not study agents' ability to compose meaningful words to denote referents, i.e. they did not address the second level of the duality of patterning.

One of the goals of emergent communication research is to develop machines that can interact with humans. As a result, a variety of referential game approaches ensure that the emergent language is as close to natural language. This can be achieved by adding a supervised image captioning objective to encourage agents to use natural language in order to solve their communicative tasks (Havrylov & Titov, 2017; Lazaridou et al., 2017). Other methods use constraints such as memory restrictions (Kottur et al., 2017) to act as an information bottleneck to increase interpretability and compositionality. While we purposefully chose a graphical sensory-motor system to ease the visualization of the emerging language, we do not inject prior knowledge or pressures to facilitate the emergence of an iconic language. Our produced utterances are completely arbitrary. This fundamentally differentiates our work from Mihai & Hare (2021b) that trains agents to communicate via sketches replicating the visual referents they name. Note also that their drawing setup does not include dynamical motor primitives and utterances are directly optimized in image space. They, moreover, allow gradients to back-propagate from listener to speaker while we use a decentralized approach. Finally, they do not consider contrastive learning. To our knowledge, CURVES is the first contrastive deep-learning algorithm successfully applied to a referential game.

There is a large body of work exploring the factors that promote compositionally in emerging languages (Kottur et al., 2017; Li & Bowling, 2019; Rodríguez Luna et al., 2020; Ren et al., 2020; Chaabouni et al., 2020; Gupta et al., 2020). In this context, a crucial question is how to actually measure it in the first place (Mu & Goodman, 2021). To this end, (Choi et al., 2018) proposes to measure communicative performances on unseen compositions of known objects as a way to evaluate compositionality. However, it has been shown that a good performance in this test may be achieved without leveraging any actual compositionality in language (Andreas, 2019; Chaabouni et al., 2020). Thus, others instead compute topographic similarities (Brighton & Kirby, 2006), measuring the correlation between distances in the utterance space (distance between signs) and distances in the referents space (such as the cosine similarity between the embeddings of objects) (Lazaridou et al., 2018). In this paper we propose to do both and study 1) the generalization to unseen combinations of abstract features and 2) topographic measures based on the Hausdorff distances between utterances denoting composition and utterances denoting isolated features.

**Contributions.** This paper introduces:

- The Graphical Referential Game (GREG): a variation of the referential game to study the formation of signs from a graphical sensory-motor system.
- CURVES: an algorithmic solution to GREG, consisting of a contrastive multimodal encoder coupled with a generative model enabling the emergence of a graphical language.
- A study of CURVES's generalization performances on compositions of features never seen during training in a simplified control setting and a more perceptually challenging one.
- A complementary analysis of the compositionality of the emerging graphical language measuring the Hausdorff distance between utterances denoting compositions and utterances denoting their constituents.
- An ablation study measuring how the motion primitives of the sensory-motor system shape the emerging symbols.

## 2 PROBLEM DEFINITION

**Graphical referential game.**   We consider a group of two agents playing a fixed number of referential games, each time alternating their roles (speaker or listener). During a game, we first present a context $R$ of $n$ objects, called referents to a speaker $S$ and a listener $L$. At the beginning of each game, the target $r^\star \in R$ is assigned to the speaker. Given this target referent $r^\star$, $S$ produces an utterance ($u$) to designate it. Based on the produced utterance $u$, $L$ selects a referent ($\hat{r}$) in $R$. The game outcome $o$ is a success if the selected referent ($\hat{r}$) matches the target $r^\star$.

**Referents.**   Referents are compositions of orthogonal vector features (one-hot vectors). Given a set of $m$ orthogonal features $F_m$, we define the set of all possible referents as $\mathcal{R}_m = \{\sum_{f \in S} f | S \subseteq F_m\}$. The subset of referents made of exactly $k$ features are thus: $\mathcal{R}_m^k = \{\sum_{f \in S} f | S \subseteq F_m, |S| = k\}$. In our experiments, we fix $m = 5$.

From these orthogonal referents, we propose to generate objects made of digit images sampled from the MNIST dataset (LeCun et al., 1998). More precisely, we define the stochastic mapping $\Phi : \mathcal{R}_m \to \tilde{\mathcal{R}}_m$ that maps each feature $f \in F_m$ to a digit class in the MNIST dataset. For each feature in a referent, we sample a random instance from the corresponding class and randomly place it on a $4 \times 4$ grid such that no number overlap. Note that the listener and speaker can perceive different realizations of $\Phi$, in this case, we say that they see different *perspectives* of the referents. More precisely, the speaker perceives the context $R$ as $\tilde{R}_S$ and its target $r^\star$ as $r_S^\star$. Similarly, the listener perceives the context $R$ as $\tilde{R}_L$ and selects a referent $\hat{r}$ among it.

We use this formalism to instantiate three settings of the Graphical Referential Game (GREG):

- *one-hot*: where referents are one-hot vectors $r \in \mathcal{R}_m$.
- *visual-shared*: where referents are MNIST digits $r \in \tilde{\mathcal{R}}_m$ and agents share the same perspective: $\tilde{R}_S = \tilde{R}_L$.
- *visual-unshared* where referents are MNIST digits $r \in \tilde{\mathcal{R}}_m$ and agents have different perspectives of referents in their contexts $\tilde{R}_S \neq \tilde{R}_L$.

**Sensory-motor drawing system.**   Utterances are produced by a sensory-motor system $M : \mathbb{R}^m \to \mathcal{U} \subset \mathbb{R}^{D \times D}$ mimicking an arm drawing sketches displayed in Figure 2(a). The arm motion is derived from Dynamical Motor Primitives (DMPs) (Schaal, 2006). The DMP is parametrized by a command vector $c \in \mathbb{R}^{20}$. It converts $c$ into a 2-dimensional drawing trajectory $T$ made of 10 coordinates $T = \{v_i\}_{i=0,\dots,9}$. This trajectory is then fed to a Differentiable Sketching model (Mihai & Hare, 2021a) generating an $D \times D$ image (in our implementation, $D = 52$). See Suppl. Section A.1 for additional implementation details of the Sensory-motor drawing system.

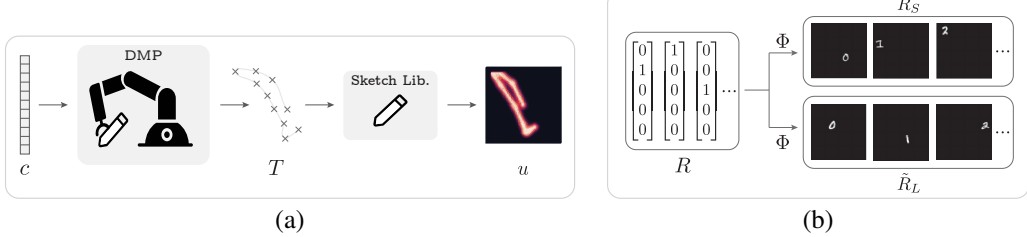

(a)                                                    (b)

**Figure 2:** (a) **Sketching sensory-motor system**: The sensory-motor system imitates a robotic arm drawing a sketch on a 2D plan. DMPs first convert a continuous command $c$ into a sequence of coordinates $T$. This trajectory is then rendered as a $52 \times 52$ graphical utterance thanks to a differentiable sketching library. (b) **Referent transformation:** An example of a one-hot context $R$ being transformed into two contexts $\tilde{R}_S$ and $\tilde{R}_L$ by the stochastic transformation $\Phi$. The two contexts are different perspectives of the same objects.

**Objectives.**   This paper investigates how a group of two agents can agree on a shared compositional language to denote referents, given a continuous sensorimotor system to produce utterances. Beyond the game's success, we evaluate the emerging language along two dimensions.

*Coherence.*   First, we measure the coherence of the emerging lexicon. As utterances are 2-dimensional paths, we compute similarities using the Hausdorff distance as it is known to capture

geometric features of trajectories, in particular, their shape (Besse et al., 2015). The Hausdorff distance $d_H$ is the maximum distance from any coordinate in a trajectory to the closest coordinate in the other: $d_H(T_1, T_2) = \max\{\sup_{v \in T_1} d(v, T_2), \sup_{v' \in T_2} d(T_1, v')\}$. In particular, we compute the following metrics.

- **Agents Coherence (A-coherence):** For a given referent $r$ with the same perspective for all agents, measure the mean pairwise similarity between each agent's utterance.
- **Perspective Coherence (P-coherence)**: For a given agent and a given referent $r$, measure the mean pairwise similarity between utterances produced from different perspectives
- **Referents Coherence (R-coherence)**: For a given agent, measure the mean pairwise similarity between utterances produced for different referents.

*Compositionality.*   The second dimension of our evaluation explores the compositional properties of the emerging language. To this end, we first evaluate the generalization performances of our group to compositional referents never seen during training. More specifically, we train agents on $\mathcal{R}_{\text{train}} = \mathcal{R}_5^1$ (referents made of one feature) and test them on $\mathcal{R}_{\text{test}} = \mathcal{R}_5^2$ (referents made of two features). For visuals about compositional referents, see Suppl. Section A.2. We use the success rate SR to monitor the performances. However, a satisfactory success rate on this testing set does not necessarily imply that the emerging language is in fact compositional (Chaabouni et al., 2020).

To complement this analysis, we estimate to what extent utterances denoting compositional referents are actually made of utterances denoting their constituents. To this end, we introduce a topographic score based on the Hausdorff distance $\rho$ that quantifies how an utterance denoting a compositional referent made of feature $i$ and $j$ ($u(r_{ij})$) is actually closer to utterances denoting isolated features $u(r_i)$ or $u(r_j)$ than the utterance naming other compositional referents ($u(r_{xy})$, $x \neq i, y \neq j$). For a detailed derivation of metric $\rho$, see Suppl. Section A.3.

## 3   CURVES - CONTRASTIVE UTTERANCE-REFERENT ASSOCIATIVE SCORING

CURVES is an energy-based approach that relies on two mechanisms:

1. The contrastive learning of an energy landscape $E(r, u)$ is defined as the cosine similarity between utterance and referent embeddings.
2. The generation of an utterance that maximizes the energy for a given target referent $r_S^\star$.

**Agents modules and interactions.**   Each agent $A \in \{A_1, A_2\}$ perceives utterances and referents using two distinct CNN encoders $f_A$ (for referents) and $g_A$ (for utterances)[1]. $f_A$ and $g_A$ map referents and utterances in a shared $d$-dimensional latent space: $f_A(\cdot, \theta_{fA}) : \mathcal{R}_m \to \mathbb{R}^d$ and $g_A(\cdot, \theta_{gA}) : \mathcal{U} \to \mathbb{R}^d$ such that $z_{rA} = f_A(r)$ and $z_{uA} = g_A(u)$, as displayed in Figure 3(a). The agent then computes the energy landscape as: $E_A(r, u) = \cos(f_A(r), g_A(u))$

A given referential game unfolds as follows. Agents have randomly attributed roles, for instance, $A_1$ is the speaker $A_1 \leftarrow S$ and $A_2$ is the listener $A_2 \leftarrow L$. The speaker is given a context $\tilde{R}_S$ and a target referent perceived as $r_S^\star$ to produce an utterance $\hat{u}$ intending to approach the utterance $u^\star$ that maximizes $E_S(r_S^\star, u)$. The listener observes $\hat{u}$ and selects referent $\hat{r}$ in context $\tilde{R}_L$ that maximizes $E_L = (r, \hat{u})$:

$$\begin{cases} \hat{u} \approx u^\star = \underset{u \in \mathcal{U}}{\operatorname{argmax}} \, E_S(r_S^\star, u) & \text{(utterance generation from speaker)} \\ \hat{r} = \underset{r \in \tilde{R}_L}{\operatorname{argmax}} \, E_L(r, \hat{u}) & \text{(referent selection from listener)} \end{cases} \quad (1)$$

The outcome of the game is then $o = \mathbb{1}_{[\hat{r}=r^\star]} - b$ where $b$ is a baseline parameter representing the mean success across previous games.

**Contrastive representation learning in referential games.**   For a given context $R$, agents are randomly assigned their roles and play $n = |R|$ games. During these $n$ games, roles are fixed and the speaker agent successively selects each referent of the context $\tilde{R}_S$ as the target $r_S^\star$. During interactions, the speaker collects data $\{(r_S^i, u^i, o^i)\}_{i=1,\ldots,n}$ while the listeners observes $\{(u^i, r_L^i)\}_{i=1,\ldots,n}$.

---

[1]when referents are one-hot vectors $f_A$ is a fully-connected network. Parameters for both encoders are given in Suppl. table 4.

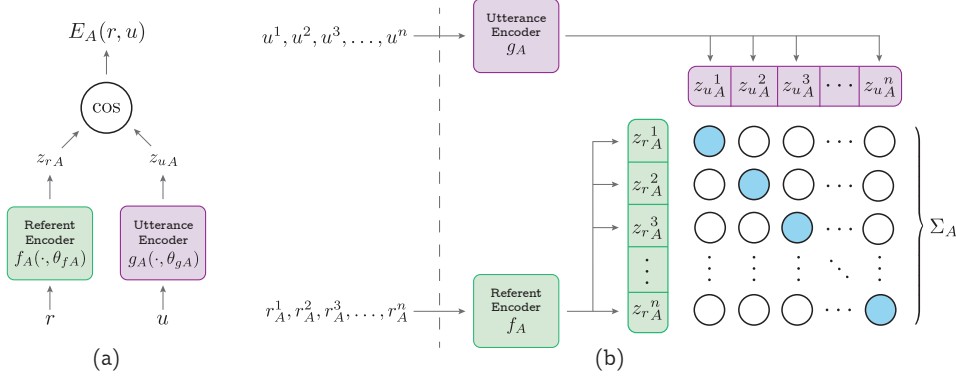

**Figure 3:** (a) **Agents's dual encoder architecture.** Referents and utterances are mapped to a share latent space. The energy between a referent $r$ and an utterance $u$ is computed as the cosine similarity between their respective embeddings. (b) **Cosine similarity matrix update from collected samples.** Agents compute the energy for all referents and utterances it collected to form the squared matrix $\Sigma_A$. During contrastive updates agents maximize blue circles and minimize white ones.

From the collected data each agent can compute the squared cosine similarity matrices $\Sigma_A$ whose elements are $(\Sigma_A)_{i,j} = E_A(r_A^i, u^j)$ as shown in Figure 3(b). Contrastive updates are then performed using the objective $J_A$ that applies *Cross Entropy* ($CE$) on the $i$-th row and $i$-th column of $\Sigma_A$.

$$J_A(\Sigma_A, i) = [CE((\Sigma_A)_{i,1:n}, e_i) + CE((\Sigma_A)_{1:n,i}, e_i)]/2, \tag{2}$$

$e_i$ being a one-hot vector of size $n$ with value 1 at index $i$. Depending on the role of the agent, $J_A$ is instantiated either as $J_S$ (speaker) or $J_L$ (listener). Thus, the speaker updates its representation using the outcomes $o_i$ of the games (reinforcing the successful associations while decreasing the unsuccessful ones):

$$\underset{\theta_{f_S}, \theta_{g_S}}{\text{minimize}} \sum_{i=1}^{n} o_i J_S(\Sigma_S, i) \tag{3}$$

On the other hand, the listener needs to make sure that the selection matches the speaker's referent (Steels, 2015) and hence always increases associations (no matter the games' outcomes):

$$\underset{\theta_{f_L}, \theta_{g_L}}{\text{minimize}} \sum_{i=1}^{n} J_L(\Sigma_L, i) \tag{4}$$

Note that in Eq. 4, $r_L^i$ is the target referent perceived by the listener. This means that the speaker indicates the referent (as perceived by the listener) that they were naming at the end of the game. This retroactive pointing mechanism was employed in both early language game implementations (Steels & Kaplan, 1999) and more recent ones (Chaabouni et al., 2020; Portelance et al., 2021).

**Speaker's utterance optimization.** We distinguish two utterance generation strategies:

- The descriptive generation: in which the speaker agent only considers the target referent $r_S^\star$ to produce an utterance that maximizes the cosine similarity between the embeddings of $r_S^\star$ and an utterance produced by our sensory system $u = M(c)$ from motor command $c$. Since $M$ is fully differentiable, we inject the sensory-motor constraint in equation 1 and seek for the optimal motor command $c^\star$ using gradient ascent:

$$c^\star = \underset{c \in \mathbb{R}^p}{\text{argmax}} \; E(r_S^\star, M(c)) \tag{5}$$

- The discriminative generation: in which the speaker also perceives the context $\tilde{R}_S$ during production. This is achieved by finding the motor command that minimizes the cross entropy given a target referent $r_S^\star$ and its context $\tilde{R}_S$:

$$c^\star = \underset{c \in \mathbb{R}^p}{\text{argmin}} \; CE(\sigma_S, e_{r_S^\star}) \tag{6}$$

where $\sigma_S$ is the vector with coordinates $\sigma_{Si} = [E(r^i, M(c))]_{r^i \in \tilde{R}_S}$ and $e_{r_S^\star}$ is the one-hot vector of size $|\tilde{R}_S|$ with value 1 at the position of $r_S^\star$ in $\tilde{R}_S$. This discriminative generation process is only used at test time when investigating CURVES's generalization capabilities.

## 4 EXPERIMENTS AND RESULTS

This section focuses first on CURVES's training dynamics as agents interact in GREG before show-casing its ability to generalize to compositional referents that were never seen during training. We, then, evaluate the compositional structure of the emerging graphical language by providing visuals of utterances and computing topographic scores defined in Section 2. Each of these studies is carried out with one-hot, shared-visual, and unshared-visual referents as explained in Section 2. We finally investigate the impact of motion primitives on the emerging language. Training and testing metrics correspond to the mean and standard deviation computed from training pairs of agents on 10 seeds.

**Do agents converge to a shared graphical language?** Figure 4 displays the training performances of a group of two agents interacting in GREG. For each referent type, the group reaches a perfect success rate of $SR = 1$. Moreover, a group starts to converge when inter-agent and inter-perspective coherence distances decrease. This correlation is proof of emergent communication as it indicates that agents start agreeing on signs to denote referents. Finally, the constant (for one-hot referent) and increasing (for visual referents) values of the R-coherence suggest that agents use distinct signs to name referents.

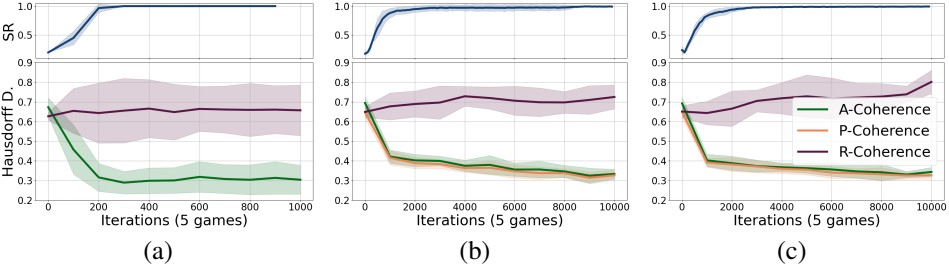

(a)  (b)  (c)

**Figure 4: Training success rate (SR) and Coherence distances** (a) one-hot referents (b) visual-shared referents (c) visual-unshared referents.

An example of an emerging lexicon describing visual referents produced by agents trained on un-shared perspectives can be visualized in Figure 5. Other visualizations for one-hot and shared visual referents are available in Suppl. Section B.2. We also provide illustrations of P-coherence in Suppl. Section B.3.

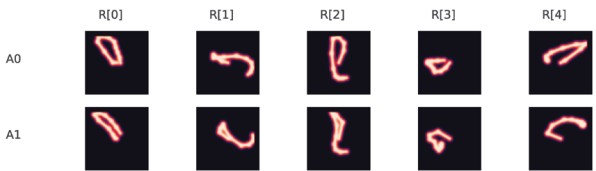

**Figure 5: Instance of an emerging lexicon.** Utterances are produced by a pair of agents trained with unshared perspectives (1 seed). The perspective for each referent is chosen randomly.

**Are agents able to generalize to compositional referents?** Table 1 exposes the generalization performances of group of agents evaluated on referents $r \in \mathcal{R}_5^2$. During an evaluation, the context is exhaustive and contains all the combinations of 2 features: $|R| = 10$. We compare the success rates to a *random* baseline where the listener always selects the referent $\hat{r}_L$ randomly no matter the utterance ($SR_{random} = 0.1$). We also introduce a *1-feature* baseline where the speaker produces an utterance $u$ that only denotes one of the two features contained in $r_S^\star$ and the listener randomly selects one of the four combinations containing the communicated feature ($SR_{1-feat} = 0.25$). The success rates for all referent types are significantly higher than the baseline values suggesting that agents are indeed able to communicate about compositional referents. Generalization performances are nearly perfect with one-hot referents but they decrease in visual settings. This performance gap can be explained by the extra difficulty of adding inter-perspective variability to the multi-agent interaction dynamic during the contrastive learning of referent representations. The better success

rates obtained in auto-learning (where a single agent plays both the speaker and the listener roles) provided in Suppl. Section B.1 seem to corroborate this hypothesis. Surprisingly, we observe that success rates for descriptive (Eq. 5) and discriminative (Eq.6) generation are very similar. This suggests that optimizing utterances so as to minimize their energy between non-targeted compositional referents ($r \in R, r \neq r^\star$) does not improve generalization performances.

| Referents | Descriptive SR | Discriminative SR |
|---|---|---|
| One-hot | $0.99 \pm 0.01$ | $0.99 \pm 0.01$ |
| Visual-shared | $0.57 \pm 0.04$ | $0.56 \pm 0.03$ |
| Visual-unshared | $0.39 \pm 0.02$ | $0.40 \pm 0.02$ |

**Table 1: Generalization performances.**    Success rates evaluated on exhaustive context $|R| = 10$ with referents $r \in \mathcal{R}_5^2$ for both generative (Eq. 5) and discriminative (Eq.6) utterance generation.

**Is the emerging language compositional?**    To investigate the compositionality of utterances we propose the topographic maps displayed in Figure 6. Each point in a topographic map is an utterance naming a compositional referent $r \in \mathcal{R}_5^2$ and has coordinate $(d_H(u(r_i), \cdot), d_H(u(r_j), \cdot))$. If utterances naming the composition of two features are indeed the compositions of the utterances used to denote each of the isolated features, we expect them to land in the bottom left of the topographic maps. Figure 6 shows that some utterances for compositional referents are indeed close in Haussdorf distance to the utterances denoting the isolated constituent features (Figure 6(b)) but others are not (Figure 6(a)).

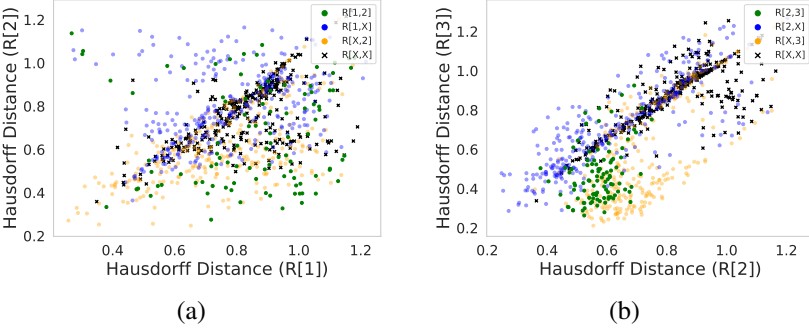

(a)                                           (b)

**Figure 6: Topographic map examples for a single seed in unshared-visual referents setting** (a) Corresponding to the worst topographic score $\rho = -0.113$ (b) Corresponding to the best topographic score $\rho = 0.203$. Each utterance names a compositional referent and is colored in blue if it contains feature $i$, orange if it contains feature $j$, green if it contains both, and black if it contains none.

Unfortunately, the feature maps do not allow us to draw strong conclusions about the composition properties of the emerging language. It is hard to tell if agents are indeed composing utterances or if the Haussdorf distance simply does not capture compositionality. This seems to be verified by additional topographic maps provided in Suppl. Section B.4. In particular, the topographic maps for one-hot referents (Figure 19) indicate that strong generalization performances can be achieved by producing utterances that are not necessarily close to the isolated feature utterances. This difficulty in evaluating compositionality can be experienced visually thanks to Figure 7 which displays a matrix of composition for unshared-visual referents. More instances of composition matrices are available in Suppl. Section B.5.

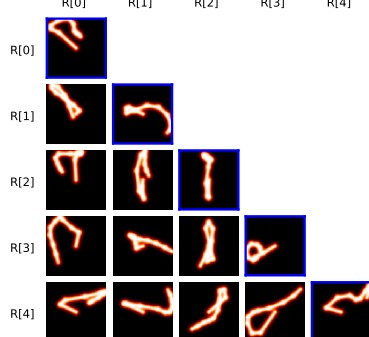

**Figure 7: Matrix of compositions.** Blue frames represent utterances generated for a perspective in $\mathcal{R}_5^1$, other utterance denote the corresponding compositions in $\mathcal{R}_5^2$

**Are representations compositional?** Finally, if compositionality is visually hard to analyze in graphical space, it seems to be much more apparent in the utterance and referent embedding spaces.

Figure 8 shows that the embeddings for compositional referents as well as the embedding of the utterances naming them are indeed close to the embeddings of their constituents. This is not surprising since this is the space in which our energy landscape is learned.

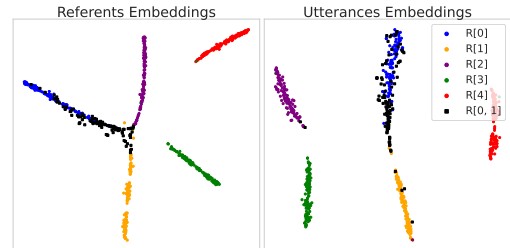

**Figure 8: T-sne of utterance and referent embeddings.** Embeddings are computed for 100 perspectives. Training conditions are unshared visual referents. Additional t-snes are provided in Suppl. Section B.6

**What is the impact of sensory-motor constraints on the emergent lexicon?** We conduct an ablation study where the speaker directly optimizes the utterance in image space (without $u = M(c)$ in Eq. 5 & Eq. 6). As reported by the training and testing performances in table 2, the pair of agents succeed during training and their generalization performances are on par with pairs producing utterances with DMPs. Crucially, the emerging lexicons, displayed in figure 9, look like noisy pixel maps and therefore have no geometrical structure. This prevents any coherence or compositionality analysis and reinforces the interest in incorporating motor constraints into the utterance generation system.

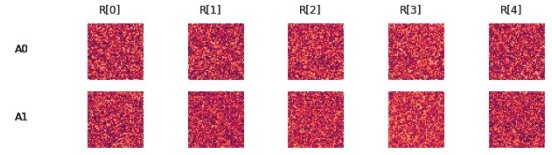

**Figure 9: Emerging lexicon without motion primitives.** Utterances naming referents with unshared perspectives.

|  | $\mathbf{SR}_{\text{train}}$ | $\mathbf{SR}_{\text{test}}$ |
|---|---|---|
| One-hot | $0.99 \pm 0.01$ | $0.96 \pm 0.02$ |
| Visual | $0.99 \pm 0.01$ | $0.41 \pm 0.02$ |

**Table 2: Training and generalization success without DMPs.** Utterances are generated in descriptive mode, and visual referents are seen from different perspectives.

## 5  DISCUSSION AND FUTURE WORK

This work formalizes GREG: a new referential game where two agents must communicate via a continuous sensory-motor system imitating a robotic arm drawing sketches. To tackle GREG, we propose CURVES: a contrastive representation learning algorithm inspired by early language game contrastive implementation that scales to high dimensional signals. CURVES allows a group of two agents two converge on a shared graphical language in contexts where referents are one-hot vectors or images of MNIST digits. The representations that agents learn enable them to communicate about compositional referents never encountered during training. Our ablation study shows that motor constraints in the utterance generation system are required to facilitate the emergence of structured lexicons. Despite the visualizable nature of graphical signs, compositions of utterances are hard to identify. Our proposed analysis based on the Hausdorff distance could not allow us to draw systematic conclusions. On the other hand, compositions are salient in the space of embeddings.

Future work may look into finding other metrics or evaluation strategies to investigate the composition of utterances in more depth. An analysis of the impact of the sensory-motor constraints on the topology of graphical signs could also provide valuable insight into the factors facilitating the emergence of a compositional graphical language. CURVES is agnostic to the modality used to represent utterances. As such, it could tackle other sensory-motor systems. The central element of CURVES lies in the contrastive learning of utterance-referent associations. In our implementation, we optmize utterances by maximizing this energy via gradient ascent. Much like CLIP opened many avenues for multi-modal generation, we could plug in more complex generative strategies such as diffusion models (Rombach et al., 2021; Saharia et al., 2022). Finally, more realistic visual referents and the impact of training larger groups of agents on generalization could be investigated in GREG.

## 6 REPRODUCIBILITY STATEMENT

We ensure the reproduciblity of the experiments presented in this work by providing our code. Additional information regarding the methods and hyper-parameters can be found in Suppl. Section A.4. Information about our derived metrics can also be found in Suppl. Section A.3. We ensure to display the variance of our experimental results by using 10 random seeds, reporting the standard deviation.

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

# SUPPLEMENTARY MATERIAL

This Supplementary Material provides additional derivations, implementation details and results. More specifically:

- Section A provides supplementary implementation details in the form of:
  - Images of testing set of visual referents;
  - Topographic score derivation;
  - Training procedures and hyperparameters;
  - Pseudo-code.
- Section B provides supplementary results:
  - Auto-comprehension generalization performances;
  - Additional Lexicons;
  - Utterances examples across perspectives illustrating coherence;
  - Topographic maps & scores;
  - Composition matrix examples;
  - T-SNEs of embeddings;

## A   SUPPLEMENTARY METHODS

### A.1   SENSORY-MOTOR SYSTEM

*Dynamic Motion Primitives.* This subsection provides additional details about the implementation of the Dynamical Movement Primitives use to produce 2-dimensional trajectories. Our drawing system consists of a 2-dimensional system that mimics the motion of a pen in a plan. Each of the $x$ and $y$ positions of the pen is controlled by a DMP starting at the center of the image and parameterized by 10 weights. These weights are the parameters of the motion of a one-dimensional oscillator that generates a smooth trajectory of 10 points. The parameters of the two DMPs are given in table 3.

| Parameter | Value |
|:---:|:---:|
| Number of weights | 10 |
| Delta time | 0.1 |
| Number of points | 10 |
| Weights range | [-500, 500] |
| Position Init. | 0 |

**Table 3:** DMP parameters for each of the two coordinate motions

*Sketching Library.* Trajectories obtained with the DMPs are then mapped to a 52x52 grid which is converted to an image with the `raster` and `softor` functions of the sketching library Mihai & Hare (2021a). The drawing thickness parameter is fixed to $1e - 2$.

## A.2 TESTING SET

Figure 10 displays examples of compositional referents made of 2 features.



Figure 10: **Perspective instances of the testing set** $\mathcal{R}_5^2$.

## A.3 TOPOGRAPHIC SCORE

To evaluate the compositionality of the emerging language we define the topographic score:

$$\rho_{ij} = ||(O, h_{ij})||_2 - ||(O, h_k)||_2 \text{ with } k = \operatorname{argmin}_{k \in \{i,j\}} ||h_k, h_{ij}||_2) \tag{7}$$

It is obtained by computing the Hausdorff distance between the utterances denoting compositional referents with respect to both the utterance denoting the single feature $i$ ($d_H(u(r_i), \cdot)$)and the one denoting the single feature $j$ ($d_H(u(r_j), \cdot)$). To derive our metric, we define 4 groups of utterances denoting compositional referents.

- $u(r_{ij})$ the utterances for referent made of feature $i$ and $j$.
- $u(r_{xj}, x \neq i)$ the utterances denoting referent made by composing feature $j$ with any other feature different than $i$
- $u(r_{iy}, y \neq j)$ the utterances denoting referent made by composing feature $i$ with any other feature different than $j$
- $u(r_{xy})$ the utterance denoting all other compositional referents in $\mathcal{R}_5^2$.

and compute their Hausdorff distances to $u(r_i)$ and $u(r_j)$. As displayed in Figure 11, if utterances $u(r_{ij})$ are compositional we expect them to be at the same time close to $u(r_i)$ and close to $u(r_j)$ and hence to land in the bottom left corner of the distance graph. Moreover, they should be closer to the origin than $u(r_{xj})$ and $u(r_{iy})$. To quantify to what extent it is the case we compute the barycenter of each group $h_i, h_j, h_{ij}$ and $h_{xy}$ and compute "how closer to the origin" is the compositional barycenter $h_{ij}$ compared to its closest barycenter using equation 7.

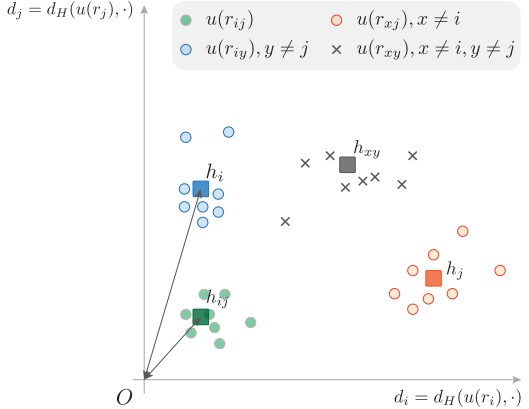

Figure 11: Idealized mapping of utterances denoting compositional referents in the plan representing distances to utterances naming isolated features $i$ and $j$.

### A.4 TRAINING PROCEDURE AND HYPERPARAMETERS

Agents have two separate encoders based on the same model architecture described in Table.4. Each agent performs association updates with a single step of gradient descent, using its own Adam optimizer with a learning rate of $1e^{-4}$. To allow faster convergence, agents perform an association update between an abstract referent $r_A^\star$ and an utterance $u$ by using a batch of 64 perspectives $\{\Phi(r_A^\star)\}_{i \in [1,64]}$. From a cognitive science perspective, this is comparable to an agent "walking around" an object to better understand how different perceptions relate to the same object. From a computer science perspective, this is similar to the self-supervised framework of SimCLR (Chen et al., 2020), where agents learn representation by contrastively aligning the embeddings of an input with these of the same transformed input.

| Layer | Activation |
|:---:|:---:|
| Conv2D(filters=8, stride=2, padding=1) | ReLU |
| Conv2D(filters=16, stride=2, padding=1) | ReLU |
| Conv2D(filters=32, stride=2, padding=0) | ReLU |
| Linear(128) | ReLU |
| Linear(32) | None |

**Table 4:**

Model architecture used for both the referent and utterance Encoders. (when referents are one-hot vectors, the 3 Conv2D layers are replaced by a Linear layer with ReLu activation)

While the drawing pipeline is fully differentiable, it is highly sensitive to local minima. Thus, we solve equation 5 in the descriptive case or equation 6 in the discriminative scenario by simultaneously performing gradient descent on a batch of $64$ randomly initialized command vectors over 100 iterations, using a newly initialized Adam optimizer each time with a learning rate of $1e^{-2}$.

## A.5 PSEUDO-CODE

---

**Algorithm 1** Speaker's Utterances

---

**Require:** perceived referents $\tilde{R}_S$, speaker's referent encoder $f_S$, speaker's utterance encoder $g_S$, sensory-motor system $M$

$Z_r \leftarrow f_S(\tilde{R}_S)$
$c \sim \text{Uniform}()$
**for** $i$ in range($N_{\text{production}}$) **do**
    $U_S \leftarrow M(c)$
    $Z_u \leftarrow g_S(U)$
    $S \leftarrow \text{sim}_{\cos}(Z_r, Z_u)$
    $\mathcal{L} \leftarrow \text{mean}(\text{diag}(S)) * (-1)$
    GD step on $c$ to minimize $\mathcal{L}$
**end for**
**Return** $M(c)$

---

---

**Algorithm 2** Listener's Selections & Binary Outcomes

---

**Require:** perceived referents $\tilde{R}_L$, produced utterances $U_S$, listener's referent encoder $f_L$, listener's utterance encoder $g_L$

$Z_r \leftarrow f_L(\tilde{R}_L)$
$Z_u \leftarrow g_L(U_S)$
$S \leftarrow \text{sim}_{\cos}(Z_r, Z_u)$
$t \leftarrow \text{argmax}(S, \text{axis}=1)$
$o \leftarrow \mathbf{0}$
**for** $i$ in range($N_{\text{referents}}$) **do**
    $o_i \leftarrow \mathbb{1}_{[t_i = i]}$
**end for**
**Return** $o$

---

---

**Algorithm 3** Agents's Association Losses

---

**Require:** perceived referents $\tilde{R}_A$, produced utterances $U_A$, outcomes $o$, agent's referent encoder $f_A$, agent's utterance encoder $g_A$

$Z_r \leftarrow f_A(\tilde{R}_A)$
$Z_u \leftarrow g_A(U_A)$
$S \leftarrow \text{sim}_{\cos}(Z_r, Z_u)$
$\mathcal{L}_0 \leftarrow CE(S, \text{reduction}=\text{False})$
$\mathcal{L}_1 \leftarrow CE(S^\top, \text{reduction}=\text{False})$
$\mathcal{L} \leftarrow (\mathcal{L}_0 + \mathcal{L}_1)/2$
**if** $A = $ "S" **then**
    $\mathcal{L} \leftarrow (\mathcal{L} \cdot o)/N_{\text{referents}}$
**else**
    $\mathcal{L} \leftarrow (\mathcal{L} \cdot \mathbf{1})/N_{\text{referents}}$
**end if**
**Return** $\mathcal{L}$

---

# B SUPPLEMENTARY RESULTS

## B.1 AUTO-COMPREHENSION GENERALIZATION PERFORMANCES

| Ref. | Auto | Social |
|---|---|---|
| One-hot | $0.997 \pm 0.005$ | $0.991 \pm 0.015$ |
| Visual-shared | $0.862 \pm 0.034$ | $0.559 \pm 0.027$ |
| Visual-unshared | $0.425 \pm 0.016$ | $0.388 \pm 0.02$ |

**Table 5:** Descriptive Success Rate

| Ref. | Auto | Social |
|---|---|---|
| One-hot | $0.997 \pm 0.005$ | $0.992 \pm 0.009$ |
| Visual-shared | $0.812 \pm 0.019$ | $0.567 \pm 0.034$ |
| Visual-unshared | $0.466 \pm 0.019$ | $0.404 \pm 0.019$ |

**Table 6:** Descriminative Success Rate

We define the **Auto** performance metric as the communicative success rate, on test set, for language games involving a single agent playing as both the speaker and listener. We compare **Auto** and **Social** performances (the latter involving pairs of different agents, as done until now) in Tables 6 & 5.

## B.2 ADDITIONAL LEXICONS

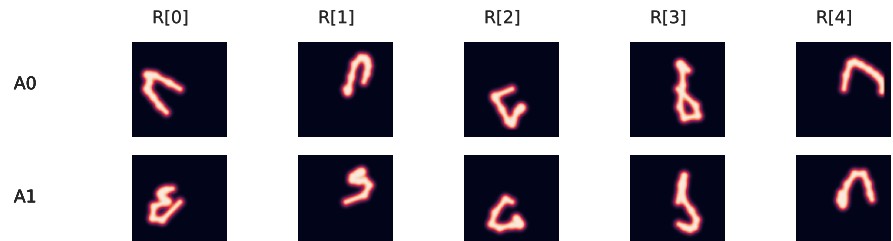

**Figure 12: Instance of an emerging lexicon.** (Visual-shared).

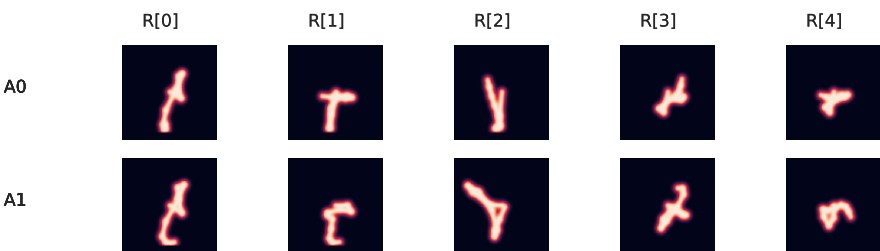

**Figure 13: Instance of an emerging lexicon.** (One-hot).

### B.3 UTTERANCES EXAMPLES ACROSS PERSPECTIVES ILLUSTRATING COHERENCE.

The following figures illustrate the P-coherence and A-coherence of an emerging lexicon (Visual-unshared) by displaying, for each referent in $R_1$, the descriptive utterance produced for 10 random perspectives.

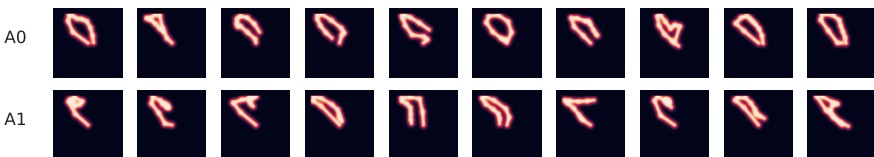

**Figure 14:** Utterances examples for referent 0.

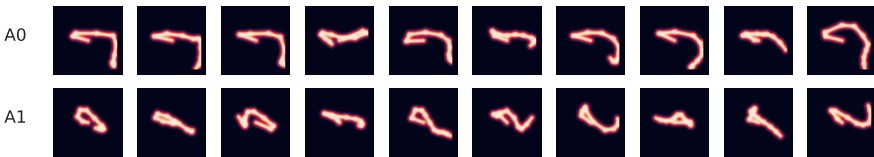

**Figure 15:** Utterances examples for referent 1.

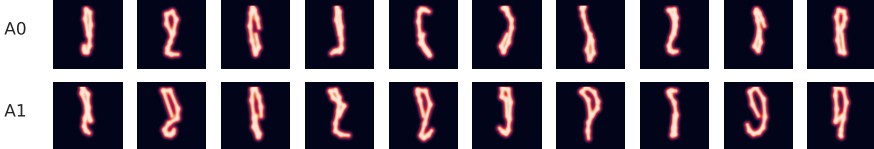

**Figure 16:** Utterances examples for referent 2.

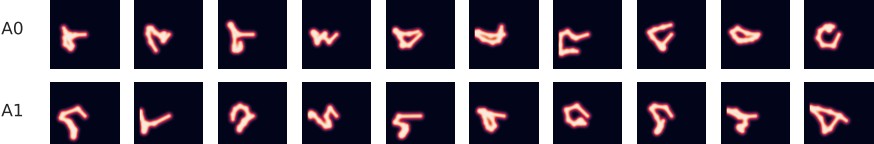

**Figure 17:** Utterances examples for referent 3.

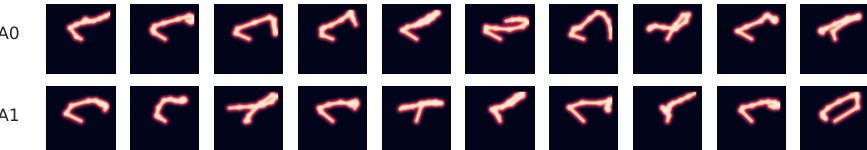

**Figure 18:** Utterances examples for referent 4.

### B.4 TOPOGRAPHIC MAPS & SCORES

### B.4.1 ONE-HOT

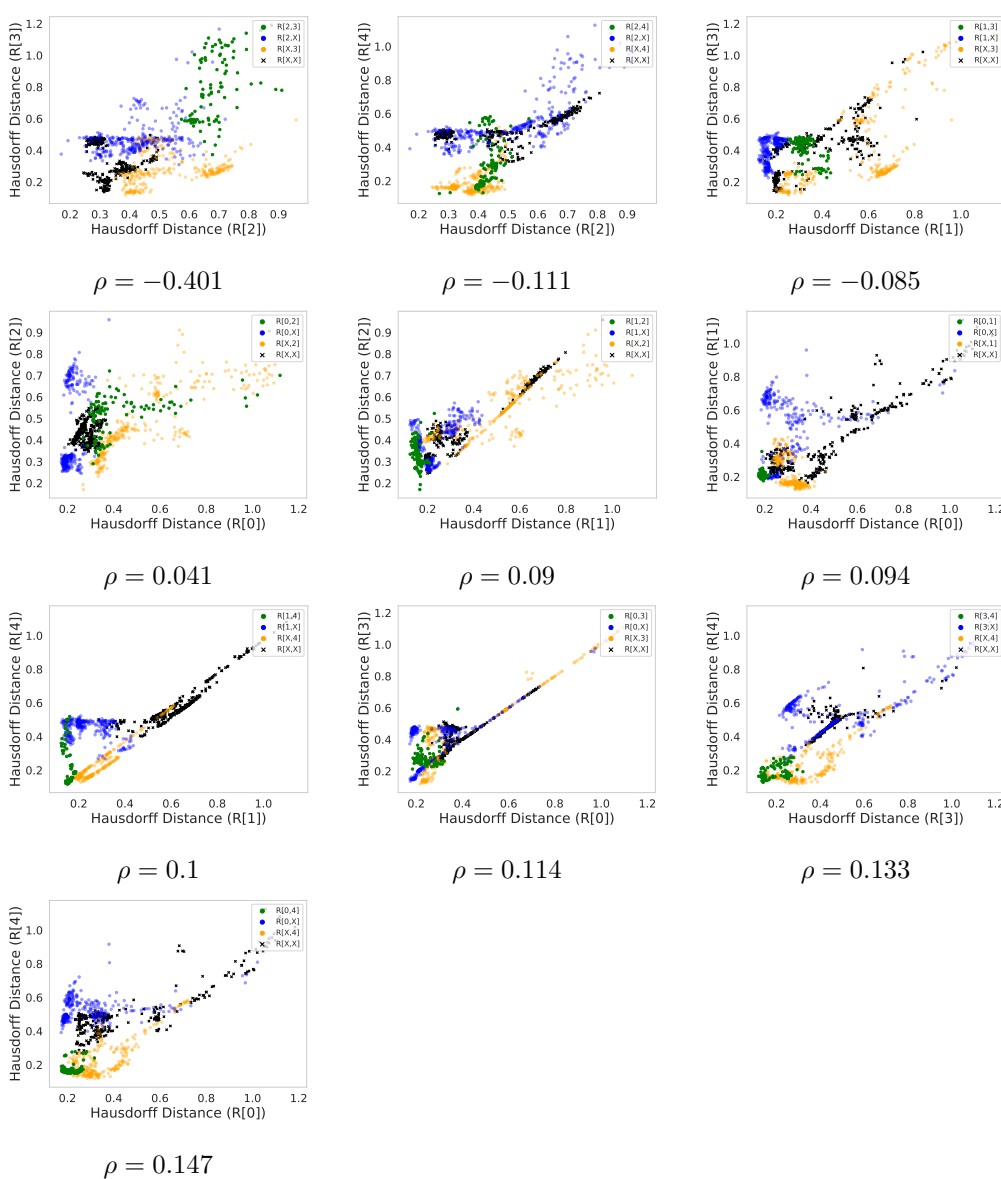

**Figure 19:** Topographic maps and their associated topographic scores for each combination of features with one-hot referents

## B.4.2 VISUAL - SHARED PERSPECTIVES

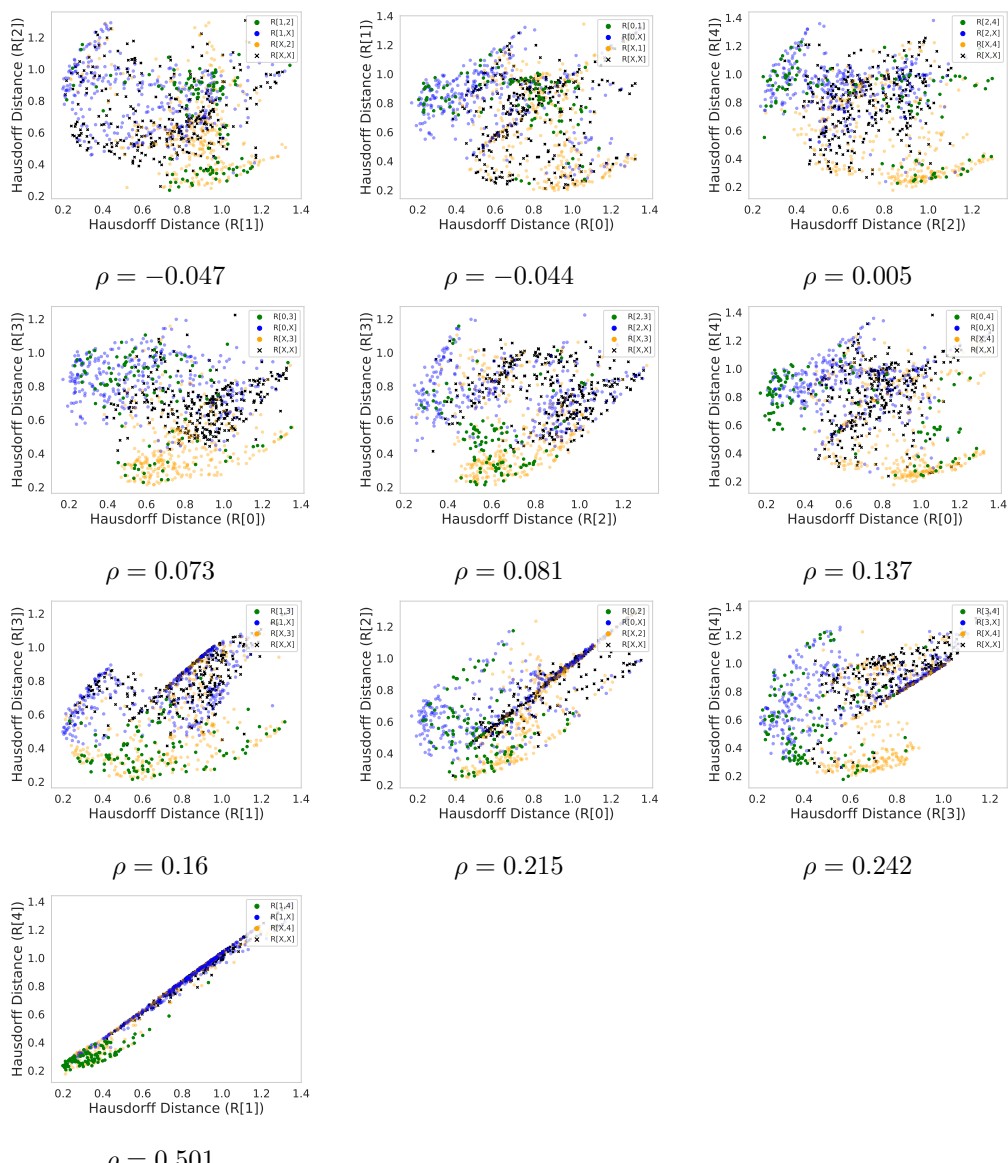

**Figure 20:** Topographic maps and their associated topographic scores for each combination of features with shared-visual referents

### B.4.3 Visual - Unshared Perspectives

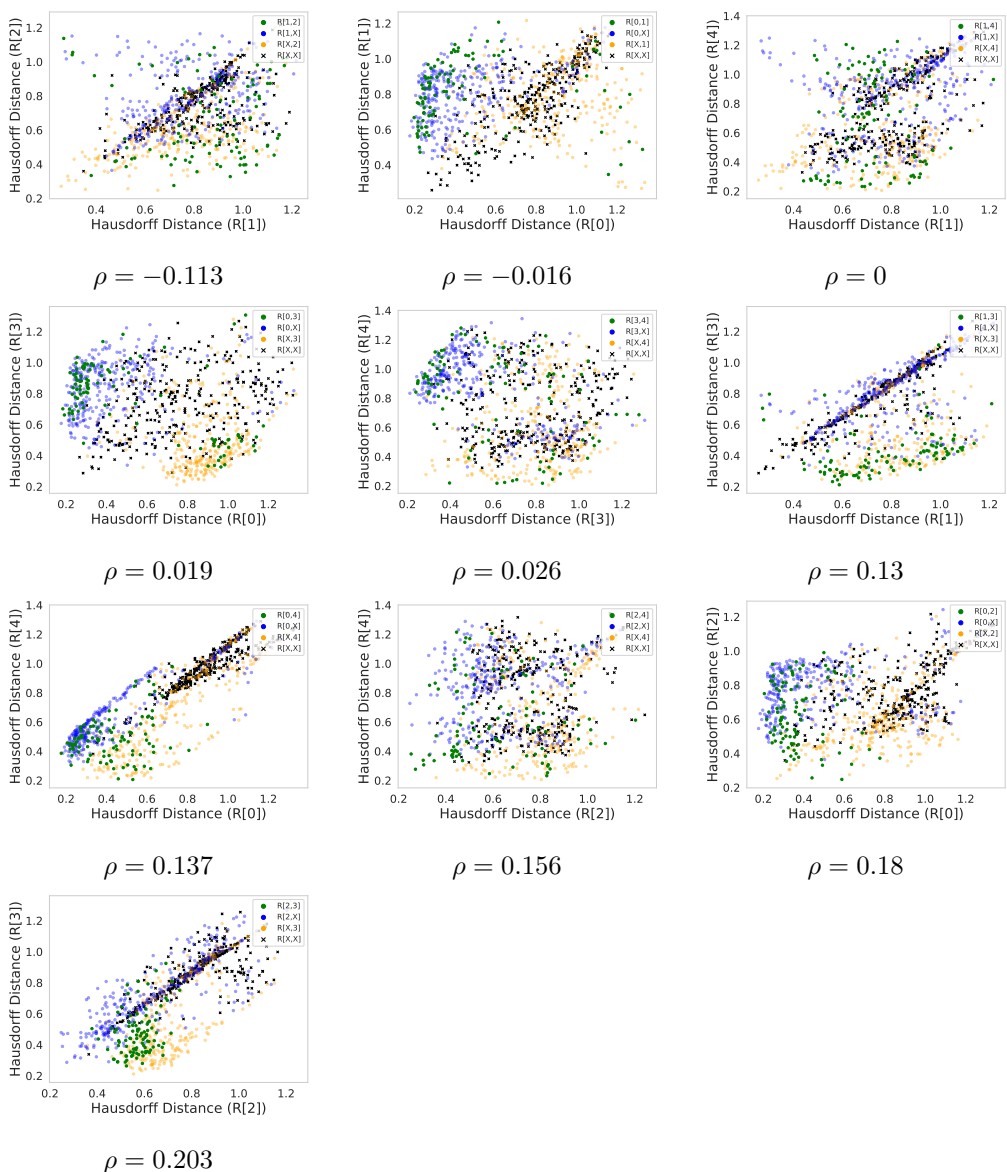

**Figure 21:** Topographic maps and their associated topographic scores for each combination of features with unshared-visual referents

### B.5 COMPOSITION MATRIX EXAMPLES (VISUAL - UNSHARED PERSPECTIVES)

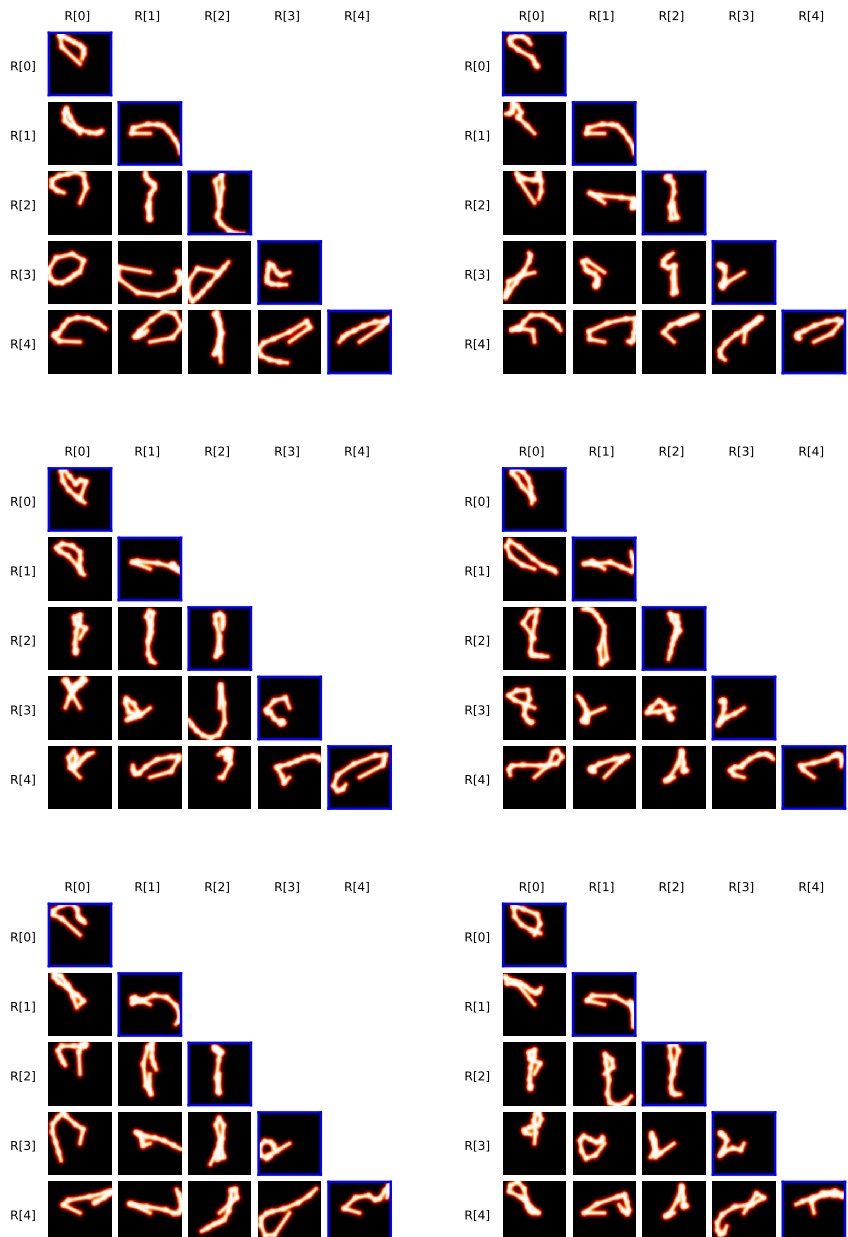

**Figure 22:** Instances of descriptive utterances for referents from $R_1$ (blue frames) and $R_2$.

## B.6 T-SNEs of embeddings (Visual - Unshared Perspectives)

### B.6.1 $R_2$ referents & descriptive utterances

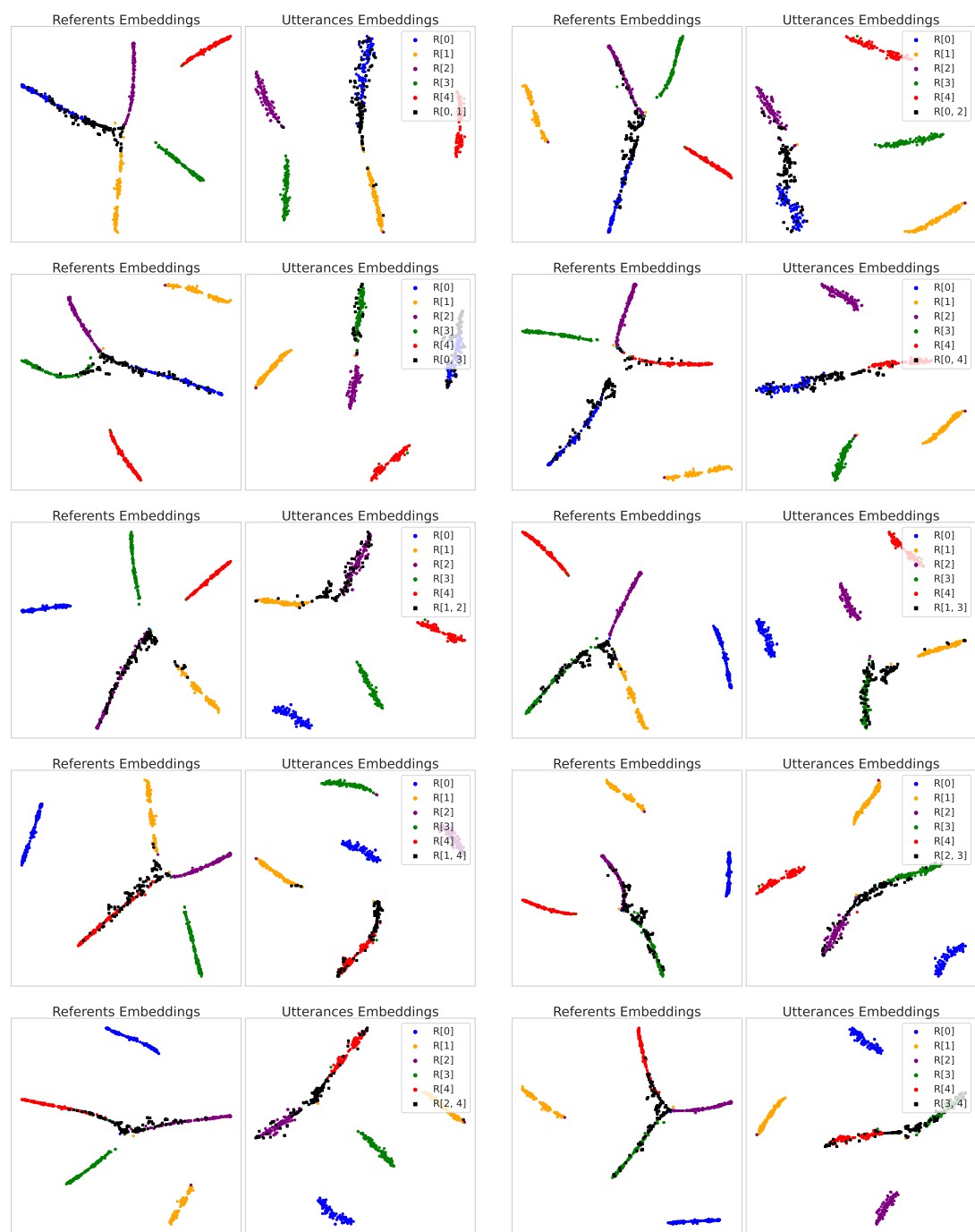

**Figure 23: T-sne of referent and descriptive utterance embeddings.** Embeddings are computed for 100 perspectives of referents from $R_2$. Training conditions are unshared visual referents.

## B.6.2 $R_2$ REFERENTS & DISCRIMINATIVE UTTERANCES

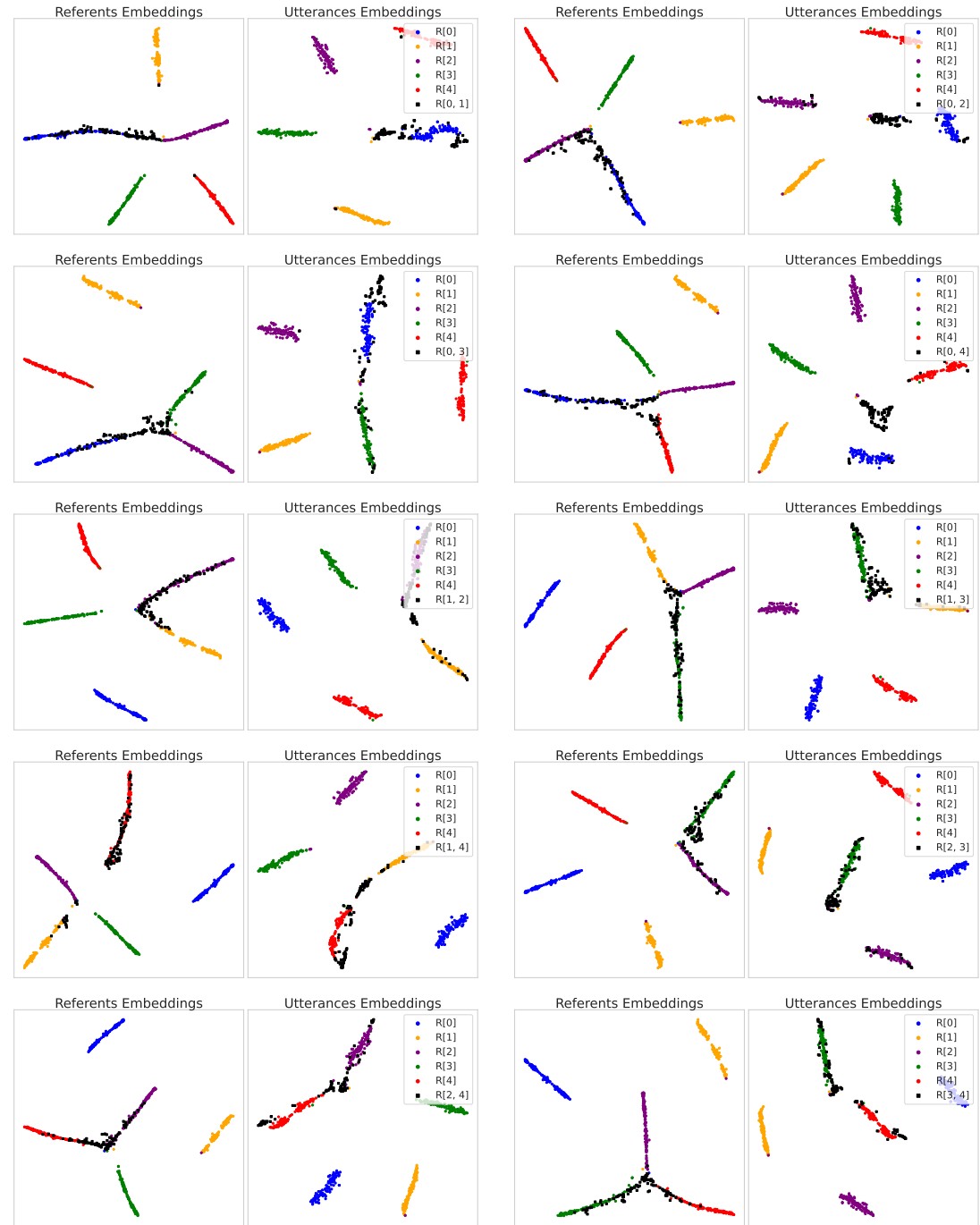

**Figure 24: T-sne of referent and discriminative utterance embeddings.** Embeddings are computed for 100 perspectives of referents from $R_2$. Training conditions are unshared visual referents.

