# OpenReview forum: "Emergence of shared sensory-motor graphical language from visual input"
_ICLR.cc/2023/Conference — Submitted to ICLR 2023_

### Official Review · Reviewer_Jswi · 2022-10-24

**Confidence:** 3
**Correctness:** 3
**Technical Novelty And Significance:** 2
**Empirical Novelty And Significance:** 2
**Recommendation:** 3

**Clarity, Quality, Novelty And Reproducibility:**

Clarity: The paper is readable, but I think clarity could be improved in several aspects. For example, there are often repetitions within the text (e.g., it is said several times that “for a given context R the agents are randomly assigned their roles”; also the objectives / contributions / scope of the paper are often redundant). The authors initially mention that agents can perceive referents “from different perspectives” without explaining what it means (the notion of “perspective” is introduced later). Furthermore, the authors discuss the distinction between discriminative vs. descriptive versions of the game, but such distinction is treated in a confused way during the presentation of the results.

Quality: The methodology seems appropriate, and the results appear robust. At the same time, I think that further analyses are required to better clarify the weaknesses mentioned above.

Novelty: Some ideas presented in this paper are original, but the overall framework seems to be partially incremental with respect to previous work (e.g., Mihai and Hare, 2021). One key innovation introduced here is the use of contrastive learning to promote a better alignment between the representations of referents and utterances; however, in my opinion the implications of this choice have not been systematically evaluated.

Reproducibility: I think the results are reproducible.

**Strength And Weaknesses:**

A strength of the paper is that it tackles a non-trivial problem by combining recent ideas from different disciplines (e.g., language games, speaker/listener communication architectures, contrastive representation learning, differentiable computer graphics…). The proposed framework is well-motivated, and the work stimulates further thinking.

At the same time, however, I see several weaknesses that might downplay the significance of the work:
- Besides an increase in ecological / cognitive realism, it is unclear what could be the advantage of using motor primitives to create utterances in the graphical space, rather than directly optimizing the continuous graphical representation (as done in previous work). Introducing a “differentiable motor system” might be relevant to better understand how sensory-motor constraints shape our ability to create written symbols, but this aspect has not been investigated in the present work.
- The authors also argue that a possible advantage of exploiting a drawing framework is to increase the interpretability of the emerging lexicon; however, also this aspect has not been emphasized in the simulations (and, in fact, the emergent signs appear quite arbitrary).
- It is not clear what is the relationship between the complexity of the concept space (e.g., similarity between objects and/or number of different objects to represent) and the emerging graphical conventions; further analyses would be needed to clarify this important aspect.
- What is the role of the dimensionality of the graphical space in the emerging lexicon? I guess this parameter might have a major impact on the type of signs that could emerge, but it seems that the dimensionality was arbitrarily chosen.
- From my understanding, the contrastive learning mechanism should (indirectly) promote an increase in similarity between the referent images and the generated utterances. However, this does not seem to happen in practice. Am I missing something?


**Summary Of The Paper:**

In this paper, the authors introduce a graphical referential game to study the emergence of visual conventions between communicating agents. The framework is based on recent computational approaches exploring the emergence of language through language games. In particular, the authors exploit a “Listener / Speaker” architecture, where the goal of the speaker is to produce an utterance that can be perceived by the listener to identify a target referent. Utterances are generated by feeding command vectors to a differentiable sketching model, with a contrastive learning objective promoting the alignment between referent and utterance embeddings. The authors also explore the compositional properties of the emerging lexicon.

**Summary Of The Review:**

I enjoyed reading this paper and I found it stimulating. At the same time, I think that the overall clarity of the manuscript should be improved, and the results should be extended to clarify several aspects of the proposed architecture. The feeling is that part of the results is more “preliminary” than ready to be published in a top-tier venue like ICLR.

---

> ### Author Response · Authors · 2022-11-12
> **Answer to Jswi**
>
> ## Main response
>
> Reviewer Jswi raises an important point and points out that beyond the introduction of the sensory-motor system that enables drawing, we do not study the aspects of the system that shape the ability to create written symbols. More precisely, they ask “what could be the advantage of using motor primitives to create utterances in the graphical space, rather than directly optimizing the continuous graphical representation”.
>
> We would like to thank reviewer Jswi for this comment which allowed us to carry out an additional experiment and propose a new contribution to the revised paper. In this new experiment, we perform the ablation suggested by reviewer Jswi and remove the dynamical motion primitives from the utterance generation system. In this scenario, the speaker directly optimizes utterances in the pixel space. We observe that although the pair of agents successfully converge to a language enabling them to solve the GREG and to generalize to OOD referents, the emerging lexicons (displayed in figure 9 of the revised paper) look like noisy images and have no apparent geometrical structure. With such utterances, it is unclear how to characterize the language. It is cumbersome to investigate the coherence or the compositionality of the emerging symbols.
>
> These observations relate to another comment of reviewer Jswi about the interpretability of the emerging lexicon: “a possible advantage of exploiting a drawing framework is to increase the interpretability of the emerging lexicon; however, also this aspect has not been emphasized in the simulations (and, in fact, the emergent signs appear quite arbitrary).” Indeed, this experiment clearly shows that even if emergent graphical lexicons are arbitrary and hard to decode by humans their structure facilitates their characterization and their analysis.
>
> Warning: The new experiment was conducted with a single seed for unshared-visual referents. Generalization results may thus vary, but extrapolating from previous experiments we expect them to have a low variance. We will update the paper with more statistically supplied results as soon as the experiments are done
>
> ## Answers to questions and comments
>
> _“From my understanding, the contrastive learning mechanism should (indirectly) promote an increase in similarity between the referent images and the generated utterances. However, this does not seem to happen in practice. Am I missing something?”_
>
> Having symbols that look like the referents would be a solution to increase the cosine similarity of their embeddings if the same encoder was used for the two modalities. However, in our implementation, we use separate encoders for utterances and referents. There is, therefore, no reason for the emerging language to resemble the referents.
>
> _“The authors initially mention that agents can perceive referents “from different perspectives” without explaining what it means (the notion of “perspective” is introduced later).”_
>
> We changed the organization of section 2 to introduce referents right after the description of the Graphical Referential Game. We hope that this change increases the clarity of the problem definition.
>
> _“Furthermore, the authors discuss the distinction between discriminative vs. descriptive versions of the game, but such distinction is treated in a confused way during the presentation of the results.”_
>
> In the revised paper, the definition of the Graphical Referential game has been simplified. We no longer distinguish between descriptive and discriminative games and assume that the speaker always sees the context. The difference is made during utterance generation: the speaker can decide to only use the target referent to generate drawings (descriptive) or also leverage distractor referents (discriminative).
>
> Information:  This response was edited to remove a buggy preliminary result.

---

> > ### Comment · Reviewer_Jswi · 2022-12-01
> > **Response to authors**
> >
> > I thank the authors for their thoughtful answers to my questions, and for seriously considering my suggestions during the revision phase.
> >
> > Regarding the ablation study, I guess that more “interpretable” representations could emerge also by simply imposing some constraints on the pixel space, for example by quantizing the pixel luminance or by encouraging the formation of localized pixel structures. However, the authors are specifically arguing that simulating motor primitives is a key factor to promote the emergence of interpretable and compositional representations. Though I sympathize with this hypothesis, I do not think that the current results allow to shed light on it.
> >
> > The previous point is also related to the issue of interpretability: though I agree that the interpretability of the lexicon reported in Fig. 9 is close to zero, I still think that the interpretability of the motor-constrained utterances has not been properly characterized/analyzed by the authors.
> >
> > Overall, I appreciate the efforts of the authors to improve the paper, but in my opinion the critical issues have not been solved.

---

> > > ### Author Response · Authors · 2022-12-09
> > > **The emergence of an interpretable code of communication is not the end goal of this paper.**
> > >
> > > We thank reviewer Jswi for reacting to our response. We agree that some pixel space constraints could yield more interpretable utterances. However, it is hard to relate these kinds of constraints to human communication, as they do not incorporate any motor commands.
> > >
> > > Crucially, we would like to stress that the emergence of an interpretable code of communication is not the end goal of this paper. Rather it aims to investigate the key mechanisms that facilitate the emergence of a communication protocol in a more human-like/ecological situation than previous work, i.e. with a sensory-motor system. This investigation requires two major contributions: 1) a new experimental setup where utterances are produced using a motor system and perceived as graphical signs; 2) a new algorithm to enable agents to self-organize a shared communication system in such a sensorimotor setting. The fact that our algorithm (CURVES) allows a pair of agents to converge on a shared graphical language (as displayed in figure 5) is a proof-of-concept that more human-like communication can emerge in more human-like setups, and our study analyzes several properties of this self-organized communication system.
> > >
> > > We hope this clarifies the objectives and context of our contributions.

---

### Official Review · Reviewer_ZH5s · 2022-10-24

**Confidence:** 4
**Correctness:** 3
**Technical Novelty And Significance:** 4
**Empirical Novelty And Significance:** 4
**Recommendation:** 6

**Clarity, Quality, Novelty And Reproducibility:**

Clarity:
* Very little detail is given as to the nature of the sensory motor model that creates the sketches. As this is an important part of the system, it would be useful to include more information about how it works exactly. The reference to Schaal (2006) and Dynamic Movement Primities seems to be a very general framework. How is it instantiated here?
* Otherwise, the paper is generally very clearly written with clear justifications and relevant citations to the literature.

Novelty:
* The proposed problem builds upon, but is significantly different from, recent studies simulating the development of language between agents.

Reproducibility:
* The authors provide their code in the supplementary material for review along with model architectures and hyperparameters in an appendix

**Strength And Weaknesses:**

Strengths:
* The proposed approach addresses an important and timely problem of agents learning to generate a shared language under realistic sensory-motor constraints. As discussed in the paper, several approaches have shown promising results without such constraints, but these are not realistic models of human language development because they lack them.
* The proposed problem formulation and framework appear to be well suited to evaluating this problem. They are simple enough to be tractable, but sophisticated enough to capture the important aspects of the problem.
* The proposed metrics seem reasonable in measuring performance
* The proposed baseline shows promising results for coherence, but not compositionality. It shows that solutions to the task are possible and that the metrics can capture the performance of solutions.
* The paper is clearly written and explained.

Weaknesses:
* The abstract states, "We show that our method allows for the emergence of a shared, graphical language with compositional properties." I'm not necessarily convinced that this is true. The proposed solution does not seem to include much in the way of compositionality. The proposed metrics seem to allow for compositionality, but it is difficult to judge the extent to which they would successfully identify a compositional language if one developed. This statement also misled me into thinking that the proposed solution was compositional, although on re-reading, it does not actually state this.

**Summary Of The Paper:**

This paper introduces a sensory-motor shared language task between two agents, metrics for evaluating the coherence and compositionality of the language that the agents develop, and an initial algorithm for solving this task using a contrastive multi-modal encoder coupled with a generative model. Experiments conducted on this solution show that the languages that develop are coherent, but not very compositional.

**Summary Of The Review:**

This paper poses an interesting and important problem and appears to provide reasonable means of evaluating solutions, although it is difficult to say for sure without an actual full solution. It does provide one solution, but it seems to fall short in terms of compositionality. The paper could provide slightly more detail on the actual sensory-motor aspect of the utterance production process, which is a crucial aspect of its novelty.

---

> ### Author Response · Authors · 2022-11-12
> **Answer to ZH5s**
>
> We thank reviewer Zh5s for their review and for pointing out the relevance of the problem we study.
>
> The main concern of reviewer Zh5s is the misleading statement about the compositional properties of emergent language that we make in the last sentence of the abstract. As we answered reviewer XbbP, we amend this sentence in the revised version of the paper.
>
> Reviewer Zh5S also suggested providing more details about the implementation of the sensory-motor system. Unfortunately, we are unable to include these details in the main part of the document as we are running out of space (due to the many changes made to the manuscript). However, we created a section in the supplementary materials (section A.1) providing extra information on our motion primitives implementation with a table including their parameters.

---

### Official Review · Reviewer_3FEJ · 2022-10-24

**Confidence:** 4
**Correctness:** 3
**Technical Novelty And Significance:** 2
**Empirical Novelty And Significance:** 3
**Recommendation:** 5

**Clarity, Quality, Novelty And Reproducibility:**

# Questions
- Some of the claims could be better supported, and the information available to each agent should either be clarified or modified (see below)
- Section 2. pg. 5 "Agents could, for instance, denote compositional referents using newly invented signs." I don't think I follow the argument here. If the agents invent new symbols at evaluation time, how can they perform better than chance? Moving to the domain of joint symbols essentially introduces a new alphabet. Is there a way to communicate referents in this alphabet on-the-fly without using compositions of existing symbols?
- The notation and terminology are sometimes difficult to follow. The use of the terms "referent", "context", and "feature" could be clarified, especially in section 2. As far as I understand it, a "referent" can refer to a single MNIST instance (caption of figure 1), but it can also refer to a collection of instances (section 2). In section 2, the term "feature" always refers to a single MNIST instance. And in the caption of figure 1, "context" always refers to a collection of instances. Have I understood correctly? I suggest disambiguating the use of the terms "referent" and "context".
- In section 2, referents $\mathcal{R}_m$ are defined terms of vectors. Why not define a referent as a set? Indeed, the notation $r\in \mathcal{R}_m$ is used in the same paragraph.
- Can I get clarification on the loss function (section 3, pg. 5-6)? I'm a little confused as to how the loss for the listener is constructed, and I have two hypotheses for what is happening. (1) The loss for the listener is optimized to learn high similarity between the uterrance embedding and the listener predicted referent. (2) The loss for the listener is optimized to learn high similarity between the utterance embedding and the ground truth referent. Do either of these match what is being described? If so, should this phrase "make sure that its selection matches the speaker’s referent" instead say "make sure that its selection matches the predicted referent"
- Along the same lines, on page 5, what is $r_L^i$? Is it the listener's prediction for the referent, or is it the ground truth label?
- Figure 4. What do the error bars indicate?
- Figure 5. What do the axis labels mean? The caption refers to a "group" of agents? Do A[0] and A[1] refer to separate groups?

**Strength And Weaknesses:**

# Strengths
- Exploring compositionality using an automatic topographic metric is an interesting idea. It would be nice to see what this metric gives for real world ideograms/pictograms to get a sense of its trustworthiness and interpretability.
- The continuous setting is a very nice departure from the typical setup

# Weaknesses
- The setting is limited. In the single-target case, the alphabet has 5 symbols. In the joint-target case, there are 5 choose 2 = 10 symbols.
- Studying compositionality in this setting is thus limited to exploring how conjunctive symbols may be encoded, as opposed to studying negation, disjunction, adjectives, adverbs, hierarchy, etc.
- In this work, compositionality is studied by looking at the form of produced symbols. For instance, to study the compound symbol $u(r_{ij})$, the authors use a topographic score to compare the compound symbol to the corresponding individual constituent symbols. However, there are many ways for a writing system to distinguish between semantics without dramatically distinguishing between forms and vice versa. It would make more sense to do this type of analysis in terms of the symbols decodability. Look at what sort of mistakes a decoding model makes. If the symbol $u(r_{ij})$ only encodes the joint symbol, and nothing about the constituent pieces, then we would expect no correlation among the incorrectly decoded symbols. But if compositionality is employed, we would expect to see predictions of the form $u(r_{ik})$ and $u(r_{ki})$ at higher frequency.
- The emergent language is not conclusively shown to be compositional (section 4) using the proposed topographic measure
- I don't think the setting can be rightly called a "language game" in which the speaker and listener are jointly learning to communicate. From section 3, pgs. 5 and 6, as best I can tell, the listener's loss is directly optimized to learn the association between the ground truth referent and the utterance. Please correct me if I'm wrong (see clarification question in section below). I believe the correct thing to do would be for the listener to learn an association between the predicted referent and the utterance. As it stands, this is information that is typically not available to the listener during the language game


**Summary Of The Paper:**

The authors study emergent machine language. They propose a game and then a system for solving the game. In the game, two agents, a speaker and a listener, must communicate the identity of a referent symbol. In the simplest version of the game, the speaker receives a target referent symbol from a set of context symbols. The speaker produces an utterance, a 2D curve in the plane, which the listener must interpret to predict the target. The authors propose a system for playing this game in which both speaker and listener use a contrastive objective to learn associations between utterances and referents. The proposed system is able to play the game better than chance. Finally, the authors propose a topographic similarity metric to study the compositionality of the emergent symbols. They find that the proposed metric makes suggestive, but not conclusive, findings about the language of their proposed system.

**Summary Of The Review:**

The proposed system solves a game in a very limited setting (5 symbols) and the proposed metric for studying compositionality does not conclusively show compositionality for the given emergent language. A revised version of the work that shows why this metric should be trusted, perhaps by giving validation on real-world symbols, would be a much stronger submission.

---

> ### Author Response · Authors · 2022-11-12
> **Answer to 3FEJ**
>
> ## Main response
>
> We thank reviewer 3FEJ for their extensive review. Their concerns mainly revolve around three dimensions: the compositionality analysis of the lexicon, the language game implementation, and the complexity of the setup. We, therefore, address these three points before moving to minor clarifications made to the paper.
>
> __Compositionality and metrics:__ Reviewer 3FEJ is questioning the choice of metric used to study the compositionality of the emergent lexicon. We would like to clarify that the metric is derived from the Hausdorff distance which is known to be efficient at capturing geometrical properties of trajectories and proven to be efficient at catching coherences for the emerging lexicon. As we discuss in the result section, we balance our analysis and report that the metric does not allow us to conclude on the compositionality of the language. In the end, we do not know whether compositionality really exists or whether we fail to measure it with our metric. We realized that the previous version of our abstract was implying a different conclusion. We thus removed this ambiguous claim. Reviewer 3FEJ proposes to evaluate the metric on human-generated pictograms to get a finer sense of its ability to grasp compositionality. This is an interesting idea but the problem would remain the same. Without knowing in advance the “composition operator” that emerges, we have no guarantee that humans would compose in the same way as the agents.
>
> __Language game implementation:__ Reviewer 3FEJ notes that “the listener's loss is directly optimized to learn the association between the ground truth referent and the utterance”  and as such the setting cannot be considered a language game. We would like to respectfully disagree with this claim. This implementation corresponds to the speaker indicating the referent (as perceived by the listener) that they were naming at the end of a game. This retroactive pointing mechanism was employed in both early language game implementations [1] and more recent ones [2,3]. We would like to use this clarification to highlight the fact that our setup is decentralized and that agents only exchange utterances (which contrasts with other implementations like [4] that share gradients).
>
> __Complexity of the setup:__ Reviewer 3FEJ mentions that our setup is limited to 5 symbols. We do agree that in our one-hot setting, we only have 5 types of referents. However, we would like to underline that we complexified this setting by adding MNIST digits with hundreds of different perspectives and random positions in a 4x4 grid. The fact that agents do not reach a 100% generalization success rate indicates that the setup is challenging enough to rigorously test our CURVES algorithm.
>
> ## Answers to questions and comments
>
> _“The notation and terminology are sometimes difficult to follow…”_
>
> We reorganized section 2 to make it clearer, simplifying the description of the game and defining referents right after the game’s description.
>
> _“In section 2, the term "feature" always refers to a single MNIST instance. And in the caption of figure 1, "context" always refers to a collection of instances. Have I understood correctly? I suggest disambiguating the use of the terms "referent" and "context".”_
>
> As mentioned in section 2., a context is a set of referents. A referent is characterized by a collection of features. A feature can be a randomly placed MNIST digit in a 4x4 grid or a value inside a vector.
>
> _“​​Section 2. pg. 5 [...]. Is there a way to communicate referents in this alphabet on-the-fly without using compositions of existing symbols?”_
>
> This claim is supported by experiments proving that “generalization to new composite concepts” does not require compositionality [2].
>
> _“Can I get clarification on the loss function (section 3, pg. 5-6)?” [...] “Along the same lines, on page 5, what is rLi? Is it the listener's prediction for the referent, or is it the ground truth label?”_
>
> We hope that the language game implementation details we provide above answer this question. To recap, rLi is the version of the target referent of the game (as perceived by the listener) or the “listener’s perspective of the ground truth”.
>
> _“Figure 4. What do the error bars indicate?”_
>
> The error bars indicate the standard deviation across different seeds.
>
>
> ## References
>
> [1] Luc Steels and Frederic Kaplan. Situated grounded word semantics. In Proceedings of the 16th International Joint Conference on Artificial Intelligence - Volume 2, IJCAI’99, pp. 862–867, SanFrancisco, CA, USA, 1999. Morgan Kaufmann Publishers Inc
>
> [2] Chaabouni et. al., Compositionality and generalization in emergent languages. In Proceedings of the 58th Annual Meeting of the Association for Computational Linguistics, pp. 4427–4442, Online, July 2020.
>
> [3] Portelance et. al. The emergence of the shape bias results from communicative efficiency. In CONLL, 2021.
>
> [4] Mihai & Hare. Differentiable drawing and sketching. NeurIPS 2021.

---

> > ### Comment · Reviewer_3FEJ · 2022-11-28
> > **Reply to authors**
> >
> > I thank the authors for their reply and answers. The authors have addressed the concern about the language game implementation in a convincing way. I will update the score from 3 to 5. The paper could be made much stronger if the emergent language could be shown to be compositional using the proposed Hausdorff distance metric.

---

### Official Review · Reviewer_5nVr · 2022-10-25

**Confidence:** 4
**Correctness:** 4
**Technical Novelty And Significance:** 2
**Empirical Novelty And Significance:** 2
**Recommendation:** 5

**Clarity, Quality, Novelty And Reproducibility:**

### Clarity
- The explanation of Greg/Curves is good and well-illustrated.
- Figure placements that break the text (e.g., Figure 4) are distracting.
  I generally recommend $\LaTeX$'s automatic figure placement.
- Figure 7 is a great visualization.
### Quality
- The experiments are a good illustration of some of the properties of the environment, but it does not seem there are any broader hypotheses about the properties of emergent language being presented or tested.
- Was Hausdorff distance the only image similarity metric you tried?
  It is a reasonable choice, but it would still be good to argue briefly as to why it is suited to this application.
### Novelty
- This paper is similar to Mihai & Hare (2021) in that it tackles emergent language with sketch-based messages, but it is sufficiently distinguished based on its unconstrained objective (no referent-utterance similarity constraint), different method of generating sketches, and different environment.
- The introduction of the new environment is not sufficiently motivated.
  Although, I see how it is different from previous environments, the paper does not argue convincingly enough for why the changes made for the environment are important in some way and not simply different.
  - From my perspective, emergent language as a field is most in need of contributions which make steps towards clear advancements and well-motivated applications.
    While in the earlier stages of the field, it was acceptable to publish more exploratory work which simply tests the waters with new environments seeing if the emergence of language was possible at all.
    But now it is pretty clear that some sort of language will always emerge and what needs to be demonstrated and discovered is how emergent languages can actually become more like human languages and how this can be used to solve real problems.
### Reproducibility
- The explanations of the method are clear, and further detail are presented in the appendix.
- Code contains dependencies and README with information on how to run an experiment.
  - Did not attempt to run code.
  - Providing dependencies with versions is even better for reproducibility (especially once the code ages for a year or five).


**Strength And Weaknesses:**

### Strengths
- (major) The paper presents a new, well-formulated emergent language environment and learning method.
- (minor) The experiments give an adequate characterization of the empirical behavior of the system.
### Weaknesses
- (major) The contributions of the paper is introducing a new environment, yet this environment is not strongly motivated.
  Simply differing from previous work is not motivation enough.
- (major) The field of emergent language does not stand to benefit that much from new environments if they do not address a relevant issue in the field.


**Summary Of The Paper:**

This paper presents a new variation of the signalling game for emergent language with 2-dimensional sketch-based messages produced via dynamical motor primitives.
To address this new environment, the authors also introduce contrastive learning-based method which aims to align the message embeddings with that of the observation in a joint space.
Finally, the paper provides an analysis of the performance, message characteristics, and compositionality of the emergent language.


**Summary Of The Review:**

I am recommending marginal reject because while I find the environment to be well-formulated, I do not see the novelties as being significant from a research perspective.
I think that given where the field of emergent language is at, the introduction of new environments needs to be focused on more tangible advancements in the field; research which is primarily exploratory is not sufficient for this.


### Misc.
- "emergent language is as close to natural language" -> add "as possible"
- "no number overlap" -> "no number overlaps"
- "that quantifies how an utterances denoting ..." -- check for typos
- Equation 1: What is $u^\star$?

---

> ### Author Response · Authors · 2022-11-12
> **Answer to 5nVr**
>
> ## Main response
>
> We thank reviewer 5nVr for their thorough feedback. We understand that their main concern is about the relevance of our proposed environment in the “field of Emergent Language”. We would, therefore, like to elaborate on this subject.
>
> We believe that our Graphical Referential Game is strongly motivated by previous works in the field that study the self-organization of language from continuous commands [1,2]. We, therefore, stand by the fact that it is an important and timely research question that was never explored with neural network agents (recent works always use discrete channels of communication). This argument is supported by reviewers Zh5s and Jswi.
>
> In the meantime, we understand reviewer 5nVr’s criticism since we did not provide any experimental evidence illustrating the interest in using a continuous sensory-motor system for the production of utterances. This is why we propose a new contribution in the revised version of the paper, which we describe hereafter.
>
> We conduct an experiment where we remove the dynamical motion primitives from the utterance generation system. In this scenario, the speaker directly optimizes utterances in the pixel space. We observe that although the pair of agents successfully converge to a language enabling them to solve the GREG and to generalize to OOD referents, the emerging lexicons (displayed in figure 9 of the revised paper) look like noisy images and have no apparent geometrical structure. With such utterances, it is unclear how to characterize the language. It is cumbersome to investigate the coherence or the compositionality of the emerging symbols. To some extent, the sensory-motor system we propose facilitates the emergence of a more human-interpretable language and addresses reviewer 5nVR’s concern about “what needs to be demonstrated and discovered is how emergent languages can actually become more like human languages”.
>
> ## Answers to questions and comments
>
> _“Was Hausdorff distance the only image similarity metric you tried? It is a reasonable choice, but it would still be good to argue briefly as to why it is suited to this application.”_
>
> We now provide justification as to why we chose the Hausdorff distance in the Objective subsection of section 2 in the new version of the paper (we added a reference to [3]).
>
> _“Equation 1: What is u⋆?”_
>
> u* is the optimal solution to the maximization problem that we try to approach using gradient ascent. We now explain it in the main text.
>
> ##References
>
> [1] Bart G. de Boer. Self-organization in vowel systems. J. Phonetics, 28:441–465, 2000
>
> [2] Pierre-Yves Oudeyer. Self-organization in the evolution of speech. In Oxford Studies in the Evolution of Language, 2006.
>
> [3] Besse et. al. Review and perspective for distance based trajectories clustering, 2015.
>
> Information: This response was edited to remove a buggy preliminary result.

---

> > ### Comment · Reviewer_5nVr · 2022-11-17
> > **Same concerns**
> >
> > Thank you for the reply.  I do think the environment is novel, but I am not
> > convinced that the environment presented is going to serve as a building block
> > in the progress of emergent language research.  2-3 years ago, I was more
> > optimistic with the introduction of new emergent language environments with the
> > hope that the accumulation of research papers would yield some general
> > knowledge and breakthrough results.  But after having seen more and more
> > environment introduced on top of the extant multitude, I tend to be skeptical
> > of the applicability of an environment without either an overwhelming intuitive
> > appeal or a concrete story of what the environment will be applied to.  In my
> > estimation, this paper does not currently satisfy either criterion.

---

> > > ### Author Response · Authors · 2022-11-18
> > > **Our environment allows us to formalize, implement and experiment with a fundamental scientific question: how can a language self-organize from sensory-motor interactions?**
> > >
> > > We thank reviewer 5nVr for reacting to our response. We would like to stress that the introduction of our environment aims at gaining fundamental knowledge about the mechanisms involved in the self-organization of a language from realistic motor commands when two artificial agents attempt to name visual referents.
> > >
> > > Sensory motor constraints are at the heart of human communication which relies on the production of motor commands by a speaker (e.g. activation of arm muscles or the vocal tract) and their perception by a listener (through vision or hearing). The complex non-linear relationship between these motor commands and their resulting perceptions is known to highly influence the way humans [1,2] or artificial agents [3] communicate, as explained in section 1 of the paper. This comes in stark contrast with all recent work in emergent communication (from the first paper of Lazaridou [4] in 2017) which has never considered such sensory-motor constraints.
> > >
> > > In this regard, the introduction of the Graphical Referential game is not intended to be applied to a downstream task but rather to address a fundamental question in emergent language that was never tackled before with deep neural network (DNN) agents. In other words, to reuse the terminology of the reviewer, we believe the “applicability” of our environment is shown in our paper through its use to formalize, implement and experiment how two artificial agents can learn to communicate using a continuous sensory-motor system. Here, “applicability” is aimed at making progress in fundamental scientific questions rather than downstream engineering tasks.
> > >
> > > Finally, introducing sensory-motor constraints in DNN emergent communication implies new challenges for the field, e.g. how to process high-dimensional communication signals in the form of images. These challenges are addressed by our CURVES algorithm. Therefore, we would like to emphasize that the environment is not the only contribution of our paper. The introduction of our CURVES algorithm and the analysis of the emerging graphical code are indeed central and innovative elements of our work. Moreover, as we mention in the discussion, CURVES is agnostic to the modality used to represent utterances and can be therefore applied to tackle a variety of sensory-motor systems.
> > >
> > > [1] Kenneth N. Steven. _On the quantal nature of speech. Journal of Phonetics_. 1989
> > >
> > > [2] C. Moulin-Frier, J. Diard, J.-L. Schwartz, P. Bessière. _Cosmo (“communicating about objects using sensory–motor operations”): A bayesian modeling framework for studying speech communication and the emergence of phonological systems_. Journal of Phonetics. 2015
> > >
> > > [3] P.Y. Oudeyer, _Self-Organization in the Evolution of Speech_. 2006
> > >
> > > [4] A. Lazaridou, A. Peysakhovich, M. Baroni, _Multi-Agent Cooperation and the Emergence of (Natural) Language_. ICLR 2017.

---

### Official Review · Reviewer_XbbP · 2022-10-26

**Confidence:** 3
**Correctness:** 3
**Technical Novelty And Significance:** 3
**Empirical Novelty And Significance:** 2
**Recommendation:** 6

**Clarity, Quality, Novelty And Reproducibility:**

The descriptions in the text are quite clear, however the presentation of data in Figure 4 is quite hard to read. There are some details around perception by the listener that I could not easily find in the text which would be essential for reproducibility.

**Strength And Weaknesses:**


*Strengths:*
- The experiments are well designed (one-hot vs visual-shared vs visual-unshared). This difference in perspective between the speaker and listener is a nice addition to this game.

- In the main text there is a balanced set of conclusions around compositionality which match the evidence: that the presence/absence of compositionality in the learned language is difficult to assess and conclude anything about. However (see weaknesses) the abstract makes a different claim.

Weaknesses:

- Despite the lack of clear evidence for compositionality in the main text, the abstract claims in the final sentence “we show that our method allows the emergence of a shared, graphical language with compositional properties”. This claim is unsubstantiated and should be removed.

- It wasn’t clear to me how the drawn image is actually perceived by the listener. Is there a CNN in here somewhere to instantiate this perception, that I am missing? If the listener’s perception happens instead in the action space of the speaker then is it really a continuous signal? The perception of the image by the listener needs to be clarified in the text.

- Compositionality is a hypothesis which is generally difficult to falsify empirically. It is not clear what data would have yielded a clear compositional vs not-compositional result either way. It is therefore not clear what this section of the paper on measuring compositionality contributes to the literature.

- Figure 4 is a bit hard to parse as it has multiple axes going on at the same time. These results would be clearer if there was another way to present them.


**Summary Of The Paper:**

This paper investigates whether agents evolve a shared language when equipped with a sensorimotor system that can produce and perceive drawings. The paper introduces a variant of the referential game, (called GREG) in which two agents must communicate via drawings. Agent A (the speaker) produces a drawing (a continuous graphical utterance) to indicate the ‘name’ of an object in a shared visual scene and communicates this to agent B. Agent B (the listener) then has to select the object indicated by agent A.   The agents switch roles between rounds. The drawings are produced by the agents involved using basic actions over a sketching environment, and so produce apparently continuous signals instead of discrete symbols (as would be the case in written language). Note that the drawings are not constrained to visually reflect the referents.

The paper then presents a way to train agents to succeed in GREG. The method (Contrastive utterance-reference associative scoring) uses contrastive learning to represent the alignment between utterances (drawings) and referents (objects in the environment), and the generation of utterances maximises energy shared with the target referent to make  the utterance as identifiable (to the speaker, not to the listener) as possible.

The paper examines performance in this game, analyses the resulting learned language using coherence and compositionality metrics and claims that the learned language demonstrates componsitional properties.


**Summary Of The Review:**

This paper proposes a variant on the referential game, which creates a new medium for communication: drawings. This results in a continuous space of output to be perceived by the listener, which bring the game setting closer to verbal communication between agents. While the game itself is interesting, the method to solve it does not seem particularly general (which is not a big drawback, just a small one) and the section on compositionality is a weak point.

---

> ### Author Response · Authors · 2022-11-12
> **Answer to XbbP**
>
> ## Main repsonse
>
> We thank reviewer XbbP for their constructive feedback. Reviewer XbbP identified the section on compositionality as the main weakness of our work.
>
> First, we agree with reviewer XbbP that the end of the abstract was overly enthusiastic about the compositional properties of the emergent language. We, therefore, amend it.
>
> Then, as mentioned by reviewer XbbP “compositionality is a hypothesis which is generally difficult to verify empirically”. To analyze it, we follow the guidelines proposed by the literature, namely looking at generalization performances and introducing a new topographic metric based on the Hausdorff distance. Unfortunately, we cannot obtain robust and strong evidence of compositionality in doing so. We do not know whether it has really emerged or whether the metrics we use (the Hausdorff distance) do not capture its expression. However, we believe that this analysis is worth including in the paper as it further characterizes the emerging language, providing a better understanding of the geometry of symbols and their relationships to each other.
>
> ## Answers to questions and comments
>
> _“It wasn’t clear to me how the drawn image is actually perceived by the listener. Is there a CNN in here somewhere to instantiate this perception, that I am missing?”_
>
> Each agent perceives utterances and referents using two distinct CNN encoders $f_A$
> (for referents) and $g_A$ (for utterances). In the revised version of the paper, we explicitly provide these details in the main text. We also point towards a supplementary section for the exact parameters of agents’ architectures.
>
> _“Figure 4 is a bit hard to parse as it has multiple axes going on at the same time. These results would be clearer if there was another way to present them.”_
>
> The new version presents a clearer version of Figure 4. Double y-labels are replaced by vertical subplots.

---

> > ### Comment · Reviewer_XbbP · 2022-11-20
> > **Thanks for the response and edits**
> >
> > Thanks to the authors for their response and to the other reviewers for their close reading. While I believe the author's changes have improved the paper, my score will remain the same.

---

### Decision · Program_Chairs · 2023-01-20

**Decision:**

Reject

**Justification For Why Not Higher Score:**

Even after revisions in response to reviewer comments, and the authors admitting that compositionality is not at all addressed either in the method or in the evaluation that the manuscript provides, the manuscript continues to make statements about compositionality.

Reviewers were unconvinced by the setting, authors did not meaningfully explain why a system for drawing sketches is a meaningful addition to the study of the emergence of language. The setting is still rather complex and could be further simplified to get at the core of the problem being addressed. Reviewers pointed out that scaling the setting up may reveal more, beyond the five symbols that are in the alphabet today, rather than focusing on a more complex output system. A similar simpler environment already exists in the literature, the manuscript should engage with that work more convincingly.

This could be a valuable contribution, but it needs more polish, precision, and direction.

**Justification For Why Not Lower Score:**

N/A

**Metareview: Summary, Strengths And Weaknesses:**

Summary: An agent must produce a drawing that allows another agent to identify a target object among N distractors.
Strengths: A novel setting for emergent language that builds on prior work to provide more flexibility while avoiding sharing gradients between speaker and listener. A novel contrastive method.
Weaknesses: The manuscript, even after edits in response to the reviewers, continues to highlight compositionality as a key feature; as the objective it lists "This paper investigates how a group of two agents can agree on a shared compositional
language to denote referents...". Having set this objective, the manuscript does not address it conclusively. Very weak evidence is provided; evidence that is far from sufficient for such a critical claim. Even the status of the resulting symbol sets as a language is unclear, as one reviewer put it "that part of the results is more “preliminary” than ready to be published in a top-tier venue like ICLR". The manuscript would be stronger if it simply never mentioned compositionality at all, as this claim is not supported. Some reviewers did not see the novel setting as a plus, questioning the need for including a drawing-based sensory system, and some questioning the complexity of the setup. Authors did simplify the setting, but more could be done here. Reviewers also pointed out that the setting is very narrow, having an alphabet of only five symbols.

**Summary Of Ac-Reviewer Meeting:**

N/A